# TO IMPUTE OR NOT TO IMPUTE? MISSING DATA IN TREATMENT EFFECT ESTIMATION

## ABSTRACT

Missing data is a systemic problem in practical scenarios that causes noise and bias when estimating treatment effects. This makes treatment effect estimation from data with missingness a particularly tricky endeavour. A key reason for this is that standard assumptions on missingness are rendered insufficient due to the presence of an additional variable, treatment, besides the individual and the outcome. Having a treatment variable introduces additional complexity with respect to *why* some variables are missing that is overlooked by previous work. In our work we identify a new missingness mechanism, which we term *mixed confounded missingness* (MCM), where some missingness *determines* treatment selection and other missingness *is determined by* treatment selection. Given MCM, we show that naively imputing all data leads to poor performing treatment effects models, as the act of imputation effectively *removes* information necessary to provide unbiased estimates. However, no imputation at all also leads to biased estimates, as missingness determined by treatment divides the population in distinct subpopulations, where estimates across these populations will be biased. Our solution is *selective* imputation, where we use insights from MCM to inform precisely which variables should be imputed and which should not. We empirically demonstrate how various learners benefit from selective imputation compared to other solutions for missing data.

## 1 INTRODUCTION

Treatment effects are arguably the most important estimand in causal inference [Pearl, 2009; Neyman, 1923; Rubin, 1974]. A reason for this is that a treatment *effect* lies at the heart of a causal question. Using causal inference, we try to more explicitly attribute effect to the treatment in question, by carefully disentangling the role of the environment. We shall explain with a running example.

> *Imagine, a job-training program to boost employment. Before sponsoring such a program, a legislative body may want to estimate its effect before widespread adoption. Using past data on the program, the body has to rely on causal methods to infer the effect of the training program before formal adoption.*

We define *effect* as the difference in outcome when applying the treatment versus not applying the treatment (or any alternative treatment for that matter). Literature on inferring (or predicting) treatment effects is largely concerned with handling *selection bias*. That is, we identify a possible difference between the treated and non-treated subpopulations of the data, seeing that treatment is rarely distributed uniformly across the population. If not accounted for, selection bias will lead to biased estimates. As such, many works focus on novel strategies to handle this bias [Johansson et al., 2016; Alaa & van der Schaar, 2017; Shalit et al., 2017; Rubin, 1974; Imbens & Rubin, 2015].

> *In the past, the program was not offered to everyone. Those with a job were less likely to be considered. As were those that had been unemployed for a long time. Perhaps age played a role; being less likely to switch careers, older people would benefit less from job-training.*

Treatment effect models are adopted in a wide range of fields, such as: medicine [Obermeyer & Emanuel, 2016; Alaa et al., 2021; Bica et al., 2021], marketing [Devriendt et al., 2018; Ascarza, 2018; Debaere et al., 2019], or even human resources [Rombaut & Guerry, 2020]. However, these methods have almost exclusively assumed that data is complete. From a practitioner's standpoint, this may not always be the case and, in reality, *data is often incomplete* [Burton & Altman, 2004; Lit-

tle et al., 2012]. When missingness is not properly accounted for when training these models, their adoption in critical environments, such as medicine, could lead to misguided decisions [García-Laencina et al., 2009; Little & Rubin, 2019]. We define two reasons why missingness may detrimentally influence treatment effects models. First, when certain covariates are missing when making a treatment decision, the fact that these variables are missing may also contribute to selection-bias.

> *Without knowledge of someone's age, it was deemed better to not offer the program. This is understandable, as the high price of the program requires some level of certainty.*

A second reason why missingness in the data may have disastrous effects on treatment effects models is that missingness may also be caused by the treatment. Missingness as a result of the treatment choice increases the difference between treatment subpopulations, even when there was no difference to begin with. Effectively, these set of covariates would be very much alike had it not been for their missingness. Adjusting on these missing variables introduces bias to the model.

> *Once an applicant has accepted their offer for job-training, the organisers require some additional information. Imagine, the program's registration process asking the applicant's current address, the job of their spouse, the amount of children they have, etc. Had they not accepted to participate in the program, they would be less likely to provide this additional information.*

The literature proposes two solutions to handle missingness: (i) either we *impute* the missing variables before subjecting the data to a learner [Rubin, 1978; 2004; Little & Rubin, 2019; Kallus et al., 2018]; or (ii) we consider a missing variable as another value, and use it directly [Mayer et al., 2020b;a; D'Agostino Jr & Rubin, 2000; Rosenbaum & Rubin, 1984]. We find that imputation effectively *removes information*— i.e. given that missingness may cause treatment-selection, it is much harder to handle selection bias when we are given only part of the total information. However, considering that missingness may be different between treatment subpopulations, it can also introduce bias where there is none to begin with. In this case, not imputing the data will *introduce bias*.

**Contribution.** We introduce (and motivate adoption of) a formal description of missingness in data used to estimate treatment effects [Neyman, 1923; Rubin, 1974]. In particular, we find that previous attempts— dating as far back as the 1980s [Rosenbaum & Rubin, 1984, Appendix B] — at formalising missingness in treatment effects are too general and allow for inaccurate descriptions of missingness and its impact. We illustrate why these descriptions are insufficient, and provide an alternative termed *mixed confounded missingness* (MCM). We argue that MCM is a general-purpose missingness mechanism distinct from well-known missingness mechansims such as *missing (completely) at random* [Little & Rubin, 2019], and a refinement of *conditional independence of treatment* [Rosenbaum & Rubin, 1984] that should be adopted to describe missingness when estimating treatment effects. Furthermore, based on the insights provided by MCM, we propose a strategy to handling missing data in treatment effects, termed selective imputation. Our approach is theoretically motivated and we provide empirical evidence of how methods benefit from this approach and demonstrate the harm when missingness is not correctly dealt with.

## 2 PRELIMINARIES

Estimating causal effects is a difficult endeavor, as it requires us to answer a *counterfactual* question. In particular, when we observe the outcome after applying a treatment on an individual, it is impossible to also observe that individual's outcome under alternative treatment [Holland, 1986]. As a treatment effect is defined as the difference between both outcomes, we are tasked with inferring an estimand which is *never observed*, which is crucially different from standard supervised learning.

One can estimate causal effects by conducting randomised controlled trials (RCTs) [Fisher, 1925; Neyman, 1923]. However, RCTs are often very expensive, and are sometimes considered unethical in a clinical setting [Hellman & Hellman, 1991; Edwards et al., 1999]. However, the alternative we consider in our work— estimating effects from observational data —comes with its own challenges.

Contrasting an RCT study, comparing the subpopulations associated with each treatment in an *observational* dataset will result in biased estimates. The reason lies in the difference between these subpopulations. If treatment is indeed not assigned randomly, but instead based on an individual's characteristics, then these characteristics are more represented in each subpopulation

as a result. If these characteristics also affect the outcome, then the outcomes become biased as the subpopulations are no longer comparable. This phenomenon is often termed *selection bias*.

**Notation.** Let $X \in \mathcal{X} \subseteq \mathbb{R}^d$ be the covariates of an individual; let the individual be treated with $W \in \{0, 1\}$; and let $Y \in \mathcal{Y} \subseteq \mathbb{R}$ be their observed outcome. Practically, $X$ could be a patient with lung-cancer; $W = 1$ could be chemo-therapy (and $W = 0$ radio-therapy); and $Y$ their tumour size after treatment. We use a subscript, $X_i$ to indicate the $i^{\text{th}}$ element in $X$, which means that $X_i \in \mathbb{R}$.

**Assumptions in causal inference.** Estimating unbiased treatment effects from observational data has received a lot of attention in recent years. One of the more popular avenues in the literature, is the potential outcomes (POs) framework of causality [Neyman, 1923; Rubin, 1974]. We define the PO of a treatment $w \in \{0, 1\}$ as $Y(w)$, where $Y(w)$ corresponds to the outcome an individual would have experienced had they been assigned treatment $W = w$. While the standard consistency assumption (see Assum. 1 below) allows us to interpret the observed outcome as the potential outcome of the observed treatment, i.e. $Y = Y(W)$; selection bias makes estimating $Y(\neg W)$ more involved. Countering selection bias is achieved by correctly adjusting for the confounders. In doing so, we make the following assumptions, standard in the PO-framework:

**Assumption 1** (Consistency)**.** *The observed outcome* $Y = Y(W) = Y(w)$ *if* $W = w$, *for* $w \in \{0, 1\}$ *and* $i = 1, 2, \ldots, N$, [1] *i.e. outcomes in the data correspond to one of the potential outcomes.*

**Assumption 2** (Ignorability)**.** *The joint distribution* $p(X, W, Y)$ *satisfies strong ignorability:* $Y(0), Y(1) \perp\!\!\!\perp W | X$, *i.e. the potential outcomes are independent of the treatment, conditioned on* $X$, *implying that there are no additional (unobserved) confounders beyond the variables in* $X$.

**Assumption 3** (Overlap)**.** *The distribution* $p(X, W, Y)$ *satisfies overlap:* $\exists \, \delta \in (0, 1)$ *s.t.* $\delta < p(W | X = x) < 1 - \delta, \forall x \in \mathcal{X}$, *i.e. each individual has a probability to receive either treatment.*

**Graphical models and causality.** Alternatively to POs, we can express causal relationships as a graphical model. In particular, a causal relationship is depicted as a directed edge in a directed acyclic graph (DAG) [Pearl, 2009], where a parent node is the cause and the child node is the effect. The ignorability assumption in Assum. 2 is sometimes illustrated in such a graphical model [Richardson & Robins, 2013]. Specifically, the ignorability assumption can be expressed as the DAG shown in Fig. 1a. Typically the influence of the treatment on the outcome is expressed as a *single world intervention graph* (SWIG): $(W)(w) \longrightarrow (Y)$ [Richardson & Robins, 2013]. We have removed this SWIG-path from our figures in order to focus our discussion on the path(s) between $X$ and $W$.

Note that the set of DAGs (i.e. the Markov equivalence class) that satisfy Assum. 2 encompasses more than just the DAG in Fig. 1a. Other DAGs in this equivalence class can just as easily respect Assum. 1 to 3, but they would make no sense. For example, reversing the arrow between $X$ and $Y$ would imply that outcome causes the covariates in the individuals, while still respecting ignorability. Instead, Fig. 1a is motivated through logical reasoning, where treatment and outcome is caused by the covariates. Throughout the remainder of this paper, we will build (and extend) heavily on Fig. 1a.

**Causal estimands.** We now arrive at our two estimands of interest: the *average treatment effect* (ATE), and the *conditional average treatment effect* (CATE). Given the notation above, we can define each estimand as follows:

**Definition 1** (ATE)**.** *The ATE is defined as the population wide difference between a treatment's potential outcomes. Mathematically, we can define the ATE as follows:* $\bar{\tau}(\mathcal{X}) := \mathbb{E}_{\mathcal{X}}[Y(1) - Y(0)]$.

**Definition 2** (CATE)**.** *The CATE is defined as a conditional difference between a treatment's potential outcomes. Mathematically, we can define the CATE as follows:* $\tau(x) := \mathbb{E}[Y(1) - Y(0) | X = x]$.

**Missingness.** In practice, a sample $X$ may be incomplete. For example, a clinician responding to an urgent trauma case may have to select treatment based on incomplete information. In this scenario, the incomplete variables are considered *missing*. To learn from these data we could consider *completing* this sample through imputation, but missingness in itself may be informative. Perhaps the clinician's decision would have been different if they had had complete information. Typically, we define three *mechanisms* to describe how a variable ended up to be missing: a first is *missing completely at random* (MCAR), where missingness in one variable is independent of the other variables, a second is *missing at random* (MAR), where the missingness in one variable may depend on

---

[1] The well-known *stable unit treatment value assumption* (SUTVA) assumes both no interference and consistency [Rubin, 1980]. The equation in our consistency assumption also implies no interference.

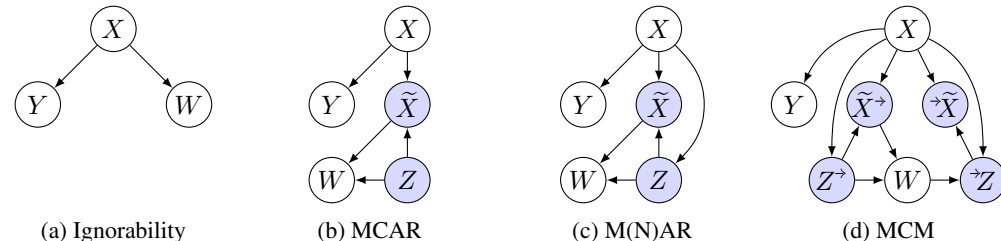

Figure 1: **[Fig. 1a] Ignorability as a graphical model.** From Richardson & Robins [2013], we express ignorability as a DAG. For brevity, we dropped the parentheses of $Y(w)$, as well as its accepted "single world intervention path" from $w$ to $Y(w)$; **[Figs. 1b to 1d] DGP for missingness mechanisms.** Shaded nodes indicate nodes that relate to missing variables, white nodes relate to treatment effects. In Fig. 1b and Fig. 1c we illustrate MCAR and M(N)AR, respectively. In Fig. 1d we illustrate MCM. Unique to MCM is to allow for treatment to cause missingness (through $\overrightarrow{\widetilde{X}}$ and $\overrightarrow{Z}$), while also allowing for treatment to be caused by missingness (corresponding to $\widetilde{X}^{\rightarrow}$ and $Z^{\rightarrow}$).

the other (observed) variables, and the third is *missing not at random* (MNAR), where missingness is typically assumed to be caused by variables outside the observed covariates [van Buuren, 2018; Rubin, 1976; Little & Rubin, 2019]. Generally, MCAR is attributed to noise in data-collection.

We indicate the missing data in $X$ with a variable $Z \in \{1, \star\}^d$, where $Z_i = \star$ if $X_i$ is missing, and $Z_i = 1$ if $X_i$ is observed. Having $Z \perp\!\!\!\perp X, Y(w)$, corresponds with MCAR; and $Z \not\!\perp\!\!\!\perp X$, but $Z \perp\!\!\!\perp Y(w)|X$, corresponds with M(N)AR. In our paper, we denote the complete (but unobserved) sample as $X$, and the incomplete (but observed) sample as $\widetilde{X} := Z \odot X \in \{\star, \mathbb{R}\}^d$ where we define $\odot$ as the element-wise product and we take $X_i \times \star = \star$, i.e. if $X_i$ is unobserved it equals $\star$ in $\widetilde{X}$.

Note that none of the existing missingness mechanisms take the treatment $W$ into account as standard, since they span a broader literature beyond treatment effects. We have illustrated MCAR and M(N)AR as DAGs in Fig. 1b and Fig. 1c, respectively, where we have included arrows from $\widetilde{X}$ and $Z$ to $W$. This corresponds to the situation where treatment is decided based on what is actually observed; the alternative where $X$ is causing $W$ would lead to a confounded setting, which is commonly assumed not to be the case in the literature. Note that these DAGs respect the conditional independent statements assumed in their respective missingness mechanisms. As we have noted in our introduction, we could either impute a missing value $X_{ij}$ with an estimate thereof, denoted $\dot{X}_{ij}$, such that we can predict a treatment effect from data with imputed samples (i.e. learn $\bar{\tau}(\dot{X})$ or $\tau(\dot{X})$); or we could predict treatment effects from data with missingness directly (i.e. learn $\bar{\tau}(\widetilde{X})$ or $\tau(\widetilde{X})$).

## 3 MIXED CONFOUNDED MISSINGNESS (MCM)

Although MCAR and M(N)AR may capture *some* settings that include a treatment, we argue that they do not capture all of them. As we will explain below, some missingness may be *caused by* treatment, while other missingness may *cause* treatment. This has important consequences as missingness is now a mixture of confounding and non-confounding elements, leading us to term our proposal *mixed confounded missingness* (MCM). We have illustrated MCM as a DAG in Fig. 1d. In this section, we will explain why there are no other arrows included in MCM, i.e. motivate that it is complete and general. Then, we compare MCM to previous proposals for missingness in CATE.

Like MCAR and M(N)AR, MCM describes the interactions between $Z$ and the remaining variables, $X, W, Y(w)$, (and $\widetilde{X}$). While MCM is fully defined by Fig. 1d, the definition rests on the assumption that $Z$ can be split into two distinct factors, $Z^{\rightarrow}$ and $\overrightarrow{Z}$, where the former captures missingness causing treatment, and the latter missingness caused by treatment. Formally, with MCM we assume:

**Definition 3** (Missingness factors in MCM). *We assume there exists a partition of $Z = \{Z^{\rightarrow}, \overrightarrow{Z}\}$ s.t. $Z^{\rightarrow} \perp\!\!\!\perp \overrightarrow{Z}|W, X$, further implying that $\widetilde{X}^{\rightarrow} \perp\!\!\!\perp \overrightarrow{\widetilde{X}}|W, X$ as $Z^{\rightarrow} \perp\!\!\!\perp \overrightarrow{\widetilde{X}}|X, W$ and $\widetilde{X}^{\rightarrow} \perp\!\!\!\perp \overrightarrow{Z}|X, W$.*

**MCM is exhaustive.** When each possible causal path is taken into account with respect to the missingness variables, we say that MCM is exhaustive. Being exhaustive is important, as through exhaustiveness can we argue that MCM covers *all* possible scenarios that can lead to missing data with treatments. Having five variables ($X, \overrightarrow{\widetilde{X}}, \widetilde{X}^{\rightarrow}, W$, and $Y$) be fully connected to $\overrightarrow{Z}$ and $Z^{\rightarrow}$ should result in 20 distinct paths (four for each of the five variables). However, we only count 6

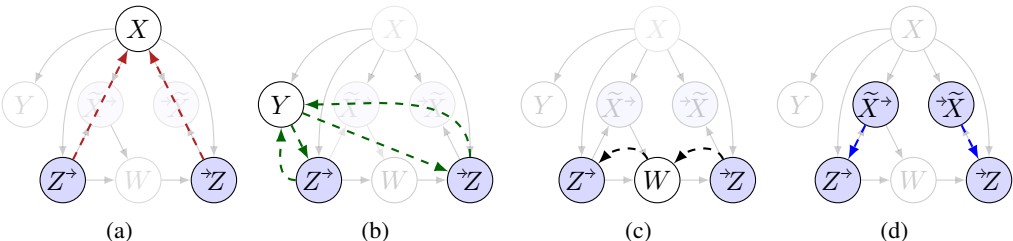

Figure 2: **Arrows that are not included in MCM.** Considering each possible direct arrow from, and to, the missingness variables ($^{\rightarrow}Z$ and $Z^{\rightarrow}$), should result in 20 arrows, as there are five remaining variables. Excluding the paths that *are* included, and the paths *across* missingness indicators, we are left with 10 paths that are *seemingly* missing from our definition. From Figs. 2a to 2d, we depict each missing arrow in function of their related variable: $X$, then $Y(w)$, $W$, and then $\widetilde{X}^{\rightarrow}$, and $^{\rightarrow}\widetilde{X}$.

direct paths in Fig. 1d that are connected to either $^{\rightarrow}Z$ or $Z^{\rightarrow}$, leaving 14 paths unaccounted for. With def. 3, we can exclude paths across missingness factors (i.e. the paths that directly connect $Z^{\rightarrow}$ and $^{\rightarrow}\widetilde{X}$, and those that directly connect $^{\rightarrow}Z$ to $\widetilde{X}^{\rightarrow}$), reducing the number of unaccounted paths to 10.

We will now argue why these 10 paths (in Fig. 2) are not included in MCM. First, we discuss Fig. 2a. Having $Z^{\rightarrow} \dashrightarrow X \dashleftarrow {}^{\rightarrow}Z$ allows covariates to change *depending on* what other variables are missing, implying that $X$ is dependent on whatever dataset an individual is included in. Having this dependence would imply that $X$, representing the fully observed– *true* –set of covariates, would be different across datasets, despite representing the same individual. Furthermore, including the paths in Fig. 2a introduces a cycle: $X \dashleftarrow\dashrightarrow {}^{\rightarrow}Z$ (and similarly for $Z^{\rightarrow}$), which violates the DAG structure.

In Fig. 2b we immediately observe cycles, meaning that only two of the four presented arrows can exist simultaneously. Let us first consider the paths where $Y(w)$ is causing missingness, i.e. $Z^{\rightarrow} \dashleftarrow Y \dashrightarrow {}^{\rightarrow}Z$. In the potential outcomes setting, $Y(w)$ is topologically last in the DAG; take Fig. 1a, where inclusion upon of the single world intervention path, $W\,w \longrightarrow Y$, we clearly see that no variable is caused by $Y(w)$. Having $Y(w)$ as the final observation makes sense; once we observe the outcome from a treatment, the covariate observations are left untouched. The absence of the remaining two arrows, $Z^{\rightarrow} \dashrightarrow Y \dashleftarrow {}^{\rightarrow}Z$, is similarly argued as the absence of the arrows in Fig. 2a. Namely, if $Z$ were to directly influence outcome, then an individual represented in two different datasets— with different missing variables —would have conflicting outcomes. Clearly, one person can only have one outcome [Holland, 1986], i.e. the arrows in Fig. 2b cannot exist.

Next, Fig. 2c, illustrating the existing arrows, *reversed*. Besides these arrows resulting in cycles— specifically, $Z^{\rightarrow} \dashleftarrow W \dashrightarrow {}^{\rightarrow}Z$ —they also violate our definition for $^{\rightarrow}Z$ and $Z^{\rightarrow}$. In particular, the *raison d'être* for these distinct factors is precisely their respective directed paths from and to $W$.

Lastly, we consider Fig. 2d depicting the reverse arrows from $^{\rightarrow}\widetilde{X}$ to $^{\rightarrow}Z$, and similarly from $\widetilde{X}^{\rightarrow}$ to $Z^{\rightarrow}$. Here too, besides the obvious cycles, $^{\rightarrow}Z \dashleftarrow\dashrightarrow {}^{\rightarrow}\widetilde{X}$ and similarly for $\widetilde{X}^{\rightarrow}$ and $Z^{\rightarrow}$, $\widetilde{X}$ is deterministically defined an element-wise product between $Z$ and $X$. This relationship is unambiguous. If indeed $^{\rightarrow}Z$ and $Z^{\rightarrow}$ are caused by the fully observed covariates $X$, it seems almost silly to consider them to be also caused by the partially observed covariates $\widetilde{X}$, which represents almost *exactly* the same entity. In fact, their only difference is completely captured in $Z$. With this, we can safely remove the arrows in Fig. 2, resulting in MCM (with a total of 6 arrows in $Z$) to be exhaustive.

**Missing data and existing assumptions.** Having argued each arrow in Fig. 1d, we will now relate MCM to previous descriptions of missingness in the treatment effects setting. Introduced in Rosenbaum & Rubin [1984] and further investigated in Mattei [2009]; Mayer et al. [2020b] and Blake et al. [2020], we find that typically the ignorability assumption in the treatment effects literature (that is, assum. 2) is extended to the setting with missingness by simply replacing the condition,

$$Y(0), Y(1) \perp\!\!\!\perp W | \tilde{X}, Z, \tag{1}$$

which translates to *"unconfoundedness despite missingness"*[2]. To verify Eq. (1), typically the above unconfoundedness assumption is combined with one of the two assumptions below [Mattei, 2009],

---

[2]Note that, we do not *have to* explicitly include $Z$ in the condition (see for example Mayer et al. [2020b]) as $\widetilde{X}$ is related in a deterministic way to $Z$. However, we chose to include $Z$ in these conditions, as they are

$$\text{CIT:} \quad W \perp\!\!\!\perp X | \tilde{X}, Z \quad \textbf{or} \quad \text{CIO:} \quad Y(0), Y(1) \perp\!\!\!\perp X | \tilde{X}, Z, \quad\quad (2,3)$$

where CIT in Eq. (2) stands for *conditional independence of treatment*, and CIO in Eq. (3) stands for *conditional independence of outcome* [Mayer et al., 2020b]. Essentially, CIT and CIO assume there is no additional information in the fully observed $X$ compared to the observed $\tilde{X}$ and $Z$ to predict treatment and outcome, respectively— i.e. adjustment for $\tilde{X}$ (and $Z$) still warrants ignorability.

The assumptions depicted in Eqs. (2) and (3) can be considered a *logical consequence* of the assumption in Eq. (1). In particular, for the potential outcomes to be ignorable from treatment, as is assumed through Eq. (1), there cannot exist direct arrows beyond those in $\tilde{X}$ from $X$ to either $W$ (which is implied by Eq. (2)), or $Y(w)$ (which is implied by Eq. (3)). Violating CIT and CIO would mean that there *has to be* such a variable (i.e. dimension) in $X$, that is not represented in the covariate space of $\tilde{X}$, that has a direct link to either $W$, or $Y(w)$, or both. Such a link is in direct conflict of Eq. (1).

At first glance, CIT and CIO may seem acceptable assumptions. However, we identify two major issues concerning these assumptions: (i) despite them further specifying the consequences of ignorability despite missingness (cfr. Eq. (1)), CIT and CIO are too general, allowing too many different (mostly completely unrealistic) missingness mechanisms without violating Eq. (1) nor CIT or CIO; and (ii) they do not allow for treatments to cause missingness, which we have argued above is an important consideration in treatment effects. Furthermore, problem (ii) hints at a larger underlying problem regarding missingness in treatment effects, which we shall discuss in Section 4.

Given the variables in Eq. (1) and CIT/CIO ($X$, $\tilde{X}$, $Z$, $Y(w)$), while keeping $\widehat{X} \rightarrow \widehat{Z} \rightarrow \widehat{\tilde{X}}$ fixed, we can generate 42 DAGs which all respect these assumptions. We have included these DAGs in Appendix D. Note that each of these 42 DAGs respects the independence statements in Eq. (1) and those in either Eq. (2) or Eq. (3). Evaluating these DAGs using the same criteria as we have for MCM leads to only 1 realistic DAG, which turns out to be a special case of MCM. Specifically, when removing the consideration of $\vec{Z}$— treatment causing missingness —we arrive at the one DAG that is realistic. Without explicit consideration of treatment choices that influence missingness (i.e. splitting $Z$ in $Z^{\rightarrow}$ and $\vec{Z}$), we learn that MCM fits nicely within CIT (and consequentially Eq. (1)). Furthermore, this one DAG corresponds exactly to how we introduced M(N)AR in Fig. 1c.

## 4 SELECTIVE IMPUTATION

Having introduced MCM, we now explain why missing data in treatment effects should not be brushed over lightly. We discuss why the two common approaches to handle missing data (recall that typically missing data is either imputed, or kept as is) are not equipped to deal appropriately with missing data that interacts with treatment. Here we offer an alternative to these two approaches.

**Why naive approaches don't work.** First, we have to discuss what can go wrong when naively dealing with missing data. To aid our discussion, we simplified MCM in Fig. 3, which merges $Z^{\rightarrow}$ into $\tilde{X}^{\rightarrow}$, and $\vec{Z}$ into $\vec{\tilde{X}}$. Note that this is equivalent to Fig. 1d as the link between the missingness indicator and the observed covariates is completely deterministic. One could, with complete certainty, derive the missingness indicator from the observed covariates. In fact, the only way of obtaining $Z$ is to do exactly that.

Imputing all data, i.e. the first naive approach, has the objective of recovering $X$ from $\tilde{X}$, as accurately as possible. Given that $Y(w)$, is a direct function of the fully observed set of covariates, $X$, and not the partially observed set, $\tilde{X}$, regressing $Y(w)$ on

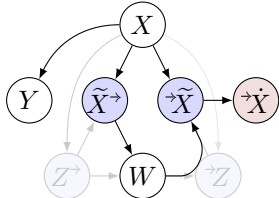

Figure 3: **Ignorability of the covariate-factors in MCM.** Above DAG depicts MCM, where we have included $Z^{\rightarrow}$ and $\vec{Z}$ in the factors $\tilde{X}^{\rightarrow}$ and $\vec{\tilde{X}}$, respectively. Besides MCM, we have also included $\vec{\dot{X}}$ as an additional node, meaning the imputed covariate set stemming directly from $\vec{\tilde{X}}$.

related to the DAG presented in Fig. 1d, where we have also included $Z$. Later in our paper, we will remove $Z$ from our DAGs and subsequently our conditions, in order to focus on the relationships with $\tilde{X}$ directly.

$\dot{X}$ should result in better estimates than $\widetilde{X}$. Contrasting supervised learning, however, the treatment effects literature is not concerned *just* with better estimates. Indeed, in treatment effects, the target is different from the *observed* outcomes in the dataset. In particular, to estimate treatment effects, one has to account for selection bias [Shalit et al., 2017], which is expressed in the path from $X$ to $W$ (see for example, Fig. 1a). In our setting, this is more complicated as there exists no direct path. The path from $X$ to $W$ is *mediated* by $\widetilde{X}^{\rightarrow}$. As explained in Section 3, treatment is decided on an individual's covariates, but also the missingness in them. As such, imputing away these missing data will result in information loss with respect to selection bias, making it all the more harder to counteract the bias when predicting a treatment effect.

Should we then impute no data, i.e. the second naive approach? Not imputing data certainly solves the problem of information loss. However, it introduces another problem. We learn from Section 3, that in order to estimate treatment effects from data, one has to assume ignorability despite missingness (see Eq. (1)). However, from Fig. 3 we learn that $Y(w) \not\perp\!\!\!\perp W | \widetilde{X}^{\rightarrow}, \overset{\rightarrow}{\widetilde{X}}$, i.e. the potential outcomes are not conditionally independent from treatment at all. We find the reason in $\overset{\rightarrow}{\widetilde{X}}$, which— due to being determined by treatment —composes a collider structure. A property of a collider structure is that is does not bias treatment selection, but when conditioned upon, *it does*. If imputing data causes problems, and not imputing data causes problems, what then should we do?

**Selective imputation.** While solutions could vary across different datasets, we provide a general strategy to handle missing data in a treatment effects setting, termed *selective imputation*. Given the aforementioned problems, it is clear that both imputing and not imputing has flaws and merits. In fact, the problem in the one, is solved by the other. Consider again Fig. 3, where we have included $\overset{\rightarrow}{\dot{X}}$ in the MCM-DAG. From evaluating Fig. 3 with this inclusion we learn that,

$$Y(0), Y(1) \perp\!\!\!\perp W | \widetilde{X}^{\rightarrow}, \overset{\rightarrow}{\dot{X}}, \tag{4}$$

arriving once more at a conditional independence between $W$ and $Y(w)$. Notice that, the independence statement in Eq. (1) conditions on the same covariate information (i.e. all is included from $X$), the only thing changed, is that we have to "forget" that some data in $\overset{\rightarrow}{\widetilde{X}}$ may be missing as the missingness associated with $\overset{\rightarrow}{\widetilde{X}}$ is determined by the treatment— a property we need to account for.

**The role of imputation.** From above discussion, we learn that missingness is what differentiates treatment-subpopulations in $\overset{\rightarrow}{\widetilde{X}}$. The following theorem shows us what exactly imputation needs to accomplish, in order to consider Eq. (4) to be true (proof of theorem 1 can be found in Appendix A).

**Theorem 1.** *Suppose the graph structure in Fig. 3, and $W$ and $\overset{\rightarrow}{\dot{X}}$ are independent, then ignorability defined as $Y(0), Y(1) \perp\!\!\!\perp W | \widetilde{X}^{\rightarrow}, \overset{\rightarrow}{\dot{X}}$ holds.*

From theorem 1 we learn that, in this context, a good imputation strategy should aim to make $\overset{\rightarrow}{\dot{X}}$ independent of the treatment, such that ignorability may hold. The intuition behind this is that through the act of (proper) imputation we effectively *balance* $\overset{\rightarrow}{\widetilde{X}}$, making the treatment populations indistinguishable from each other [Shalit et al., 2017]. The better the imputation is, the closer to $X$ the covariates in $\overset{\rightarrow}{\widetilde{X}}$ will become, reducing influence from $W$. If our imputation is of poor quality, a model may recover information about the originally missing variables, i.e. information on treatment is retained as $\overset{\rightarrow}{\dot{X}}$ still correlates with $W$, which allows bias to creep into our models.

## 5 EXPERIMENTS

We turn now to empirically validating that selective imputation, i.e. imputing only *some* parts of the covariate space, results in better treatment effect predictions. Note that good prediction performance on the outcomes present in the dataset is not considered *better* in the treatment effects literature [Alaa & van der Schaar, 2017; Shalit et al., 2017; Johansson et al., 2016], which we explain in detail below.

**Data.** Common in the treatment effects literature is the use of synthetic datasets. The reason why we have to rely on synthetic data lies at the core of the problem: in any real-world setting, *the counterfactual is unobserved*. If the counterfactual is indeed unobserved, we cannot evaluate treatment effects models on how well they predict the counterfactual. As such, with synthetic data we can simulate both potential outcomes, effectively observing the counterfactual for evaluation.

Table 1: **ATE results.** For each setting we report an MSE between the predicted ATE and the simulated (ground truth) ATE on 10 different train and test sets using 10 *differently sampled* simulations and averaged the results, standard deviation is in brackets. Imputing only $\overset{\rightarrow}{\widetilde{X}}$ ("Selective"), consistently performs best across different ATE prediction methodologies and treatments. **Lower is better**, our proposal is shaded.

| | **Impute** | | **T-Learner** | **Doubly Rob.** | **X-Learner** |
|---|---|---|---|---|---|
| | description | covariates | | | |
| **MSE** | All | $\{\widetilde{X}^{\rightarrow}, \overset{\rightarrow}{\widetilde{X}}\}$ | 0.0951 (.010) | 0.0651 (.008) | 0.0472 (.009) |
| | Nothing | $\{\emptyset\}$ | 0.0642 (.027) | 0.0902 (.018) | 0.0726 (.024) |
| | Selective | $\{\overset{\rightarrow}{\widetilde{X}}\}$ | **0.0403** (.014) | **0.0381** (.009) | **0.0309** (.014) |
| | Sel. Complement | $\{\widetilde{X}^{\rightarrow}\}$ | 0.0931 (.026) | 0.0902 (.019) | 0.0984 (.040) |

Next to counterfactual evaluation, there is an additional reason why we have to rely on a simulation: we have to be able to control the missingness mechanism. A simulation is the only way to test whether treatment effects models are affected by MCM. Our simulation is described in Appendix B.

For each experiment, we sample 10 different simulated datasets, from which we sample 10 random train and test sets, for each treatment effects method. As our finding holds both for ATE as well as CATE (in finite settings [Alaa & van der Schaar, 2018]), we test on both scenarios. Each simulated dataset contains 10k samples. The datasets span 20 dimensions (with factors of equal size), and a missingness rate of 0.3. We have performed multiple ablations on these values in Appendix C.

**Imputation.** We define four different scenarios: either we impute missing variables across (i) all the variables (indicated as "All"); (ii) none of the variables ("Nothing"); (iii) only the variables in $\overset{\rightarrow}{\widetilde{X}}$ ("Selective"); or (iv) only the variables in $\widetilde{X}^{\rightarrow}$ ("Sel. Complement"). Following Section 4, scenario (iii) (imputing only $\overset{\rightarrow}{\widetilde{X}}$) should yield the best results given that bias is removed, while information with respect to treatment selection is retained. Imputation is performed using MICE [van Buuren & Groothuis-Oudshoorn, 2011].

**Models.** In our experiments, we evaluate the performance of three classes of learners in the imputation scenarios described above: T-learner, Doubly robust (DR) learner, and X-learner. We pair each treatment effects method with `XGBoost` due to its ability to naturally handle missing values. We refer to Künzel et al. [2019] or Curth & van der Schaar [2021] for an overview of various learners.

## 5.1 AVERAGE TREATMENT EFFECTS (ATE)

**Objective.** A first estimand to consider is the average treatment effect (defined in def. 1). While the ATE has been considered since the 1970's [Rubin, 1974], it is still an important causal estimand today. It can be argued that ATEs suffer more from selection bias than the conditional ATE, as the average is computed over the *entire* dataset, unlike the CATE [Alaa & van der Schaar, 2018]. In ATE, adjustment plays an important role and is usually achieved by placing non-uniform weights on each element in the dataset when computing an average. These weights often take the form of *inverse propensity weights* (IPW), $(p(W = 1|X)^{-1})$, used by, for example, a DR-learner and X-learner.

**Results.** Consider Table 1 where we reported the mean squared error (MSE) between an estimated ATE and the ground-truth ATE across ten folds, for ten differently sampled simulations (thus spanning 100 trained learners of each type). Given three popular ATE estimation strategies, we find that imputing $\overset{\rightarrow}{\widetilde{X}}$ *significantly* performs better across all methods, confirming the insights in Section 4.

## 5.2 CONDITIONAL AVERAGE TREATMENT EFFECTS (CATE)

**Objective.** Similar to Section 5.1, we will now evaluate how CATE-learners react to MCM. Specifically, we subject the same treatment effects learners to the same scenarios as we have in Section 5.1, and evaluate their CATE-predictions using the PEHE metric described above. As was the case for our ATE experiments, the chosen learners represent a wide range of different methodologies.

Table 2: **CATE results.** We report a PEHE on 10 different random train and test sets using 10 *differently sampled* simulations and averaged the results, standard deviation is in brackets. Imputing only $\vec{\widetilde{X}}$, while keeping $\widetilde{X}^{\rightarrow}$ as is (marked as "Selective" below), consistently performs best across learners and treatments. **Lower is better**, our proposal is shaded .

| | Impute | | T-learner | Doubly rob. | X-learner |
|---|---|---|---|---|---|
| | description | covariates | | | |
| **PEHE** | All | $\{\widetilde{X}^{\rightarrow}, \vec{\widetilde{X}}\}$ | 0.7603 (0.051) | 1.3674 (1.731) | 0.6149 (0.063) |
| | Nothing | $\{\emptyset\}$ | 0.6906 (0.072) | 0.9409 (1.943) | 0.3027 (0.085) |
| | Selective | $\{\vec{\widetilde{X}}\}$ | **0.4605** (0.045) | **0.2042** (0.224) | **0.2116** (0.032) |
| | Sel. Complement | $\{\widetilde{X}^{\rightarrow}\}$ | 0.9158 (0.064) | 4.3657 (8.823) | 0.4912 (0.109) |
| **PEHE$_{W=0}$** | All | $\{\widetilde{X}^{\rightarrow}, \vec{\widetilde{X}}\}$ | 0.7371 (0.081) | 1.2083 (1.610) | 0.6272 (0.070) |
| | Nothing | $\{\emptyset\}$ | 0.7015 (0.100) | 0.8130 (1.287) | 0.2907 (0.107) |
| | Selective | $\{\vec{\widetilde{X}}\}$ | **0.5720** (0.079) | **0.1787** (0.202) | **0.2556** (0.062) |
| | Sel. Complement | $\{\widetilde{X}^{\rightarrow}\}$ | 0.9351 (0.120) | 4.2306 (9.056) | 0.5198 (0.156) |
| **PEHE$_{W=1}$** | All | $\{\widetilde{X}^{\rightarrow}, \vec{\widetilde{X}}\}$ | 0.7726 (0.055) | 1.4419 (1.802) | 0.6097 (0.068) |
| | Nothing | $\{\emptyset\}$ | 0.6881 (0.090) | 0.9973 (2.317) | 0.3091 (0.098) |
| | Selective | $\{\vec{\widetilde{X}}\}$ | **0.4097** (0.045) | **0.2169** (0.236) | **0.1915** (0.036) |
| | Sel. Complement | $\{\widetilde{X}^{\rightarrow}\}$ | 0.9183 (0.075) | 4.4322 (8.749) | 0.4803 (0.130) |

Different from evaluation in ATE, the typical evaluation metric in CATE is the *precision in estimating heterogeneous treatment effects* (PEHE), defined as, $\mathbb{E}_{\mathcal{X}}[(\tau(X) - \hat{\tau}(X))^2]$ in Hill [2011], where $\hat{\tau}$ is a model's prediction. Naturally, there is a parallel with the MSE we used in Section 5.1, where PEHE essentially corresponds to the MSE over the predicted vector of CATEs in a holdout test-set.

**Results.** Reported in Table 2 we find results for various CATE-learners across ten differently sampled simulations, each evaluated with a ten different train and test sets— as we have for our ATE in Section 5.1. Given these results, we empirically confirm that one should impute cautiously as imputing all data, no data, or wrong data, consistently performs worse than what we suggest: impute only $\vec{\widetilde{X}}$. More settings and configurations are reported in Appendix C, all of which confirm our findings.

## 6  DISCUSSION

Estimating treatment effects is becoming more important in many practical settings. The adoption of these methods is largely the result of great academic effort to further push the boundaries of these methods' abilities. While practical adoption is indeed a point in favour of causal methods, it comes with a significant downside: unlike in academia, datasets used in practice are often victim to many imperfections. In our paper, we investigated one imperfection in particular: missing variables.

Missing variables in treatment effect settings behave differently from other settings; *they should be treated differently as a result*. In our paper, we argue that some missingness can be informative of the treatment, i.e. just like certain characteristics of an individual may determine their treatment, so too can the absence of certain measurments determine treatment. If one should impute these variables, that information is lost and can no longer be used to counter any resulting selection-bias.

We argue further that we should not leave *all* variables unimputed. When there are variables that are only missing *because* an individual was given a particular treatment, then these missing variables are informative of the treatment and result in a covariate shift between treatment groups.

We believe the key finding of our paper is summarised as follows: *more care and thought should be put into imputing missing data when estimating treatment effects*. We confirm intuitively, theoretically, and empirically that selectively imputing missing variables can improve our ability to estimate treatment effect, while *wrong imputation can lead to poorly modelled treatment effects*.

## REPRODUCIBILITY STATEMENT

All our code is provide in the supplementary materials; most of it is based on open-source code such as `causalml`, `econml`, and `sklearn`. GAIN is implemented using the authors' online code (cfr. https://github.com/jsyoon0823/GAIN from Yoon et al. [2018]). Code for our data-generating process is shown in Appendix B, and is provided as separate files in the supplementary materials. We provide notebooks to reproduce the results reported in the main text as well as those reported in Appendix C.

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

## A Proof of Theorem 1

**Theorem 1.** *Suppose the graph structure in Fig. 3, and $W$ and $\vec{\dot{X}}$ are independent, then ignorability defined as $Y(0), Y(1) \perp\!\!\!\perp W | \tilde{X}^{\rightarrow}, \vec{\dot{X}}$ holds.*

*Proof.* Consider $p(Y = y, W = w | \widetilde{X}^{\rightarrow} = \tilde{x}, \vec{\dot{X}} = \dot{x})$, which we write as $p(y, w | \tilde{x}, \dot{x})$. Then by introducing $X = x$ and using the definition of the conditional density, we have

$$p(y, w | \tilde{x}, \dot{x}) = \int_x p(y, w, x | \tilde{x}, \dot{x})$$
$$= \int_x \frac{p(y, w, x, \tilde{x}, \dot{x})}{p(\tilde{x}, \dot{x})} \tag{5}$$

We can rewrite $p(y, w, x, \tilde{x}, \dot{x})$ using the graph structure as

$$p(y, w, x, \tilde{x}, \dot{x}) = p(y|x)p(\tilde{x}|x)p(w|\tilde{x})p(\dot{x}|x, w)p(x). \tag{6}$$

Noting that $X$ is *not* a collider of $W$ and $\vec{\dot{X}}$, and by independence of $W$ and $\vec{\dot{X}}$ we have that

$$p(\dot{x}|x, w) = p(\dot{x}|x) \tag{7}$$

By Lemma 1, $\widetilde{X}^{\rightarrow}$ and $\vec{\dot{X}}$ are conditionally independent given $X$. Therefore $p(\tilde{x}|x)p(\dot{x}|x) = p(\tilde{x}, \dot{x}|x)$. Similarly, $X$ d-separates $Y$ and the pair $(\widetilde{X}^{\rightarrow}, \vec{\dot{X}})$ and thus,

$$p(y|x)p(\tilde{x}|x)p(\dot{x}|x) = p(y, \tilde{x}, \dot{x}|x). \tag{8}$$

Substituting (7) and (8) into (6), gives us

$$p(y, w, x, \tilde{x}, \dot{x}) = p(y, \tilde{x}, \dot{x}|x)p(w|\tilde{x})p(x). \tag{9}$$

We then note that $\widetilde{X}^{\rightarrow}$ is *not* a collider of $W$ and $\vec{\dot{X}}$ so that,

$$p(w|\tilde{x}) = p(w|\tilde{x}, \dot{x}). \tag{10}$$

Substituting this all into (5) gives us

$$p(y, w | \tilde{x}, \dot{x}) = \int_x \frac{p(y, \tilde{x}, \dot{x}|x)p(w|\tilde{x}, \dot{x})p(x)}{p(\tilde{x}, \dot{x})}$$
$$= \frac{p(y, \tilde{x}, \dot{x})p(w|\tilde{x}, \dot{x})}{p(\tilde{x}, \dot{x})} \tag{11}$$
$$= p(y|\tilde{x}, \dot{x})p(w|\tilde{x}, \dot{x})$$

which proves the result.

$\square$

**Lemma 1.** *Suppose the setting of Theorem 1. Then $\widetilde{X}^{\rightarrow}$ and $\dot{X}$ are conditionally independent given $X$.*

*Proof.* By appealing to the DAG to write the full joint distribution of $p(x, \tilde{x}^{\rightarrow}, w, \vec{\tilde{x}}, \dot{x})$ we have

$$p(\tilde{x}^{\rightarrow}, \dot{x}|x) = \int_{\vec{\tilde{x}}, w} p(\tilde{x}^{\rightarrow}|x)p(w|\tilde{x}^{\rightarrow}, x)p(\vec{\tilde{x}}|x, w)p(\dot{x}|\vec{\tilde{x}}, x, w)$$
$$= \int_w p(\tilde{x}^{\rightarrow}|x)p(w|\tilde{x}^{\rightarrow})p(\dot{x}|x) \tag{12}$$
$$= p(\tilde{x}^{\rightarrow}|x)p(\dot{x}|x) \int_w p(w|\tilde{x}^{\rightarrow})$$
$$= p(\tilde{x}^{\rightarrow}|x)p(\dot{x}|x)$$

$\square$

## B  DATA

Standard in treatment effects literature, proper empirical validation requires synthetic data. Here we describe our synthetic setup, and how we sample from it precisely (clarifying with Python code).

**[Step 1]** *Sample $X$.* In order for imputation to make sense, there has to exist some correlation between variables in $X$, as such sampling from a standard normal with the identity matrix as a covariance matrix, will not suffice. As such, we sample from a normal distribution with a random (positive semidefinite) covariance matrix, spanning 20 dimensions:

```
1  import numpy as np
2
3  def _generate_covariates(d, n):
4      assert 0 < d
5      assert 0 < n
6
7      A = np.random.rand(d,d)
8      cov = np.dot(A, A.transpose())
9
10     X = np.random.multivariate_normal(np.zeros(d), cov, size=n)
11     X /= (X.max() - X.min())
12
13     return X
```

In above code we have `d=20` the dimension count, and `n=10000` the sample size. In our experiments we sample 10 different train and test sets from the data to calculate per simulation descriptives.

**[Step 2]** *Sample $\vec{Z}$.* From $X$ we generate $\vec{Z}$ as follows:

```
14     highest_border = X[:,:z_dim].argsort(axis=1)[:,
15         -int(np.max((int(np.round(amount_of_missingness * z_dim)), 1)))]
16     Z_out = list(x >= x[highest_border[i]] for i, x in enumerate(X[:,:
       z_dim]))
17     Z_out = np.array(Z_out).astype(int)
18     Z_out = np.abs(Z_out-1)
```

In above code we have two main parameters: `z_dim=10` indicating the amount of variables in $\vec{Z}$, and `amount_of_missingness=0.3` indicating the fraction of the data that is missing. in `ln 14` above, we calculate a threshold, when the value of $X_i$ above this threshold, that variable is missing. This threshold corresponds to the amount of dimensions that need to be missing in order to respect `amount_of_missingness`. Notice that $\vec{Z}$ now corresponds to `X[:,:z_dim]`.

**[Step 3]** *Sample $W$.* We make very explicit that treatment choices depend on $\vec{Z}$.

```
19     W = []
20     for z_d in Z_out:
21         if 0 == z_d[-1]:
22             w = 0
23         elif 0 in z_d[:int(np.floor(z_dim/2))]:
24             w = 1
25         else:
26             w = np.random.binomial(1, .5)
27         W.append(w)
28     W = np.array(W)
```

**[Step 4]** *Sample $\vec{Z}$.* Sampling $\vec{Z}$, which depends on $W$ in such a way that the arrow is identifiable, requires interaction between $X$ and $W$[3]. For this we sample two random vectors (`theta_z_in_0`, and `theta_z_in_1`) and let those interact with $X$ to decide $\vec{Z}$. As was the case with $\vec{Z}$, we also calculate the amount of variables that should be affected (calculated in `ln 29-32`). Note that the `dim_count` in this setting only corresponds *approximately* to the `amount_of_missingness`.

```
29     import scipy
30
```

---

[3]Note that in the non-binary setting, this interaction may not be necessary.

```
31      dim_count = np.round(amount_of_missingness * (d - z_dim) * 2)
32      dim_count = np.max((dim_count, 1))
33      dim_count = np.min((dim_count, int((d - z_dim) / 2)))
34      dim_count = int(dim_count)
35
36      theta_z_in_0 = np.full(dim_count, scipy.stats.norm.ppf(1 -
        amount_of_missingness))
37      theta_z_in_1 = np.full(dim_count, scipy.stats.norm.ppf(1 -
        amount_of_missingness))
38
39      Z_in = np.zeros((n, d - z_dim))
40      for i, z in enumerate(Z_in):
41          x = X[i, z_dim:z_dim+dim_count]
42          if W[i]:
43              Z_in[i, -dim_count:] = (x - X[:, z_dim:z_dim+dim_count].mean(
        axis=0)) > (theta_z_in_1 * x.std(axis=0))
44          else:
45              Z_in[i, :dim_count] = (x - X[:, z_dim:z_dim+dim_count].mean(
        axis=0)) > (theta_z_in_0 * x.std(axis=0))
46      Z_in = np.abs(Z_in-1)
```

**[Step 5] Sample $Y(w)$.** Generating outcomes is done simply by sampling two linear functions.

```
47      def _generate_outcomes(X, W):
48          theta_y0 = np.random.rand(X.shape[1])
49          theta_y1 = np.random.rand(X.shape[1])
50
51          Y0 = np.sum(X * theta_y0, 1)
52          Y1 = np.sum(X * theta_y1, 1)
53
54          Y = np.array([Y0[i] if w == 0 else Y1[i] for i, w in enumerate(W)
        ]) + np.random.randn(X.shape[0])*.1
55
56          CATE = Y1 - Y0
57
58          return Y0, Y1, CATE, Y
```

**[Step 6] Identifiability.** All the above are simple functions, which are made identifiable (such that they respect the DAG) through a non-linearity. Then, $\widetilde{X}$ is generated by combining $Z$ into $X$.

```
59      X = np.abs(X)
60
61      X_tilde = X.copy()
62      X_tilde[:,z_dim:][Z_out==0] = missing_value
63      X_tilde[:,:z_dim][Z_in==0] = missing_value
```

## C ADDITIONAL EXPERIMENTS

Consider Table 3, where we have repeated the experiment in Table 2 with GAIN [Yoon et al., 2018]; a deep learning imputation method based on GANs [Goodfellow et al., 2014]. While notoriously difficult to train, we find the combination of GAIN and MCM to be an interesting one. Specifically, the discriminator is tasked with classifying imputed samples from real samples, making it harder to predict which samples were imputed. The way the discriminator does this, is by predicting a binary mask of sorts, corresponding to what we have defined as $Z$. Having 50/50 prediction (resulting in the worst cross-entropy for binary prediction), we say that $Z$ is independent of the imputed result, which then corresponds with theorem 1. Note that in theorem 1 we pose independence of $W$. Seeing that $\vec{Z}$ is a mediator variable between $W$ and $\vec{\widetilde{X}}$, we have $\vec{\widetilde{X}} \perp\!\!\!\perp W|X$ given that $\vec{\widetilde{X}} \perp\!\!\!\perp Z|X$.

As a second additional experiment, consider Fig. 4, where we have repeated the same experiment in Table 2, but with different levels of missingness. Not only do these results confirm that selective imputation is best across the board, they also bring to light an interesting phenomenon: the trade-off between prediction-gain from imputation versus bias reduction. As $Y(w)$ is a function of the fully

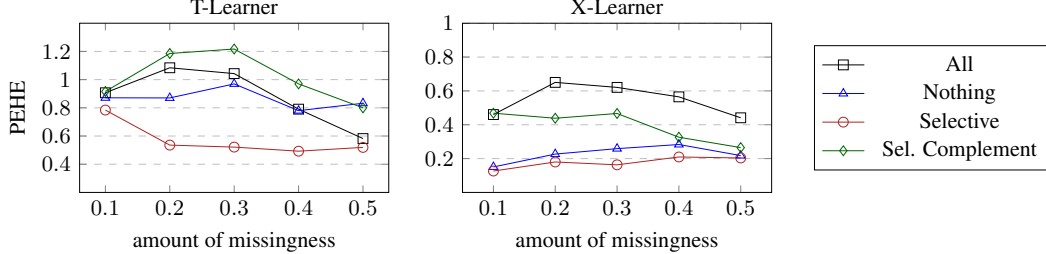

Figure 4: **Sensitivity analysis on the amount of missingness: CATE.** We report for two learners their performance (PEHE, y-axis) as a function of the amount of missingness (x-axis). For each setting of the amount of missingness, we sample 50 different train and test sets to calculate an average and std (in line with what is reported in Table 2 and documented in our provided code-files).

observed $X$, it makes sense that predicting $Y(w)$ from $X$ directly would yield the best results. This insight is the motivation why imputation is so common in standard supervised learning. From our discussion in Section 4, we learn that imputation in the treatment effects setting introduces bias, but perhaps there is only very little bias. In some cases, it might be better to impute, *despite bias*.

What we see in Fig. 4 is the combination of the balance trade-off described above, with another realisation: if too much data is missing, bias might actually reduce, as the associated patterns in the data become less apparent. Take the most extreme case where all data in $\overset{\rightarrow}{\widetilde{X}}$ is missing. Of course, then there is no bias anymore, as there is no more data to bias. With less bias in the data, it might be best for a model to impute. This is clearly visible in the leftmost plot (T-Learner), where not imputing does not seem to improve with more missingness; only the "Sel. Complement" and "All"— which introduce bias —seem to improve as more data is missing, likely since there is less bias available to introduce by imputing. Overall, we find that "Selective" outperforms the rest.

Table 3: **CATE results with GAIN.** We repeat our experiment form the main text. For each setting we report a PEHE on 100 *differently sampled* simulations and averaged the results, standard deviation is reported in brackets. Imputing only $\overset{\rightarrow}{\widetilde{X}}$, while keeping $\widetilde{X}^{\rightarrow}$ as is, consistently performs best across learners and treatments. **Lower is better**, our proposal is shaded .

|  |  | **Impute** |  | **T-learner** | **Doubly rob.** | **X-learner** |
|---|---|---|---|---|---|---|
|  |  | description | covariates |  |  |  |
| **PEHE** |  | All | $\{\widetilde{X}^{\rightarrow}, \overset{\rightarrow}{\widetilde{X}}\}$ | 0.8720 (0.072) | 1.4276 (1.410) | 0.5388 (0.112) |
|  |  | Nothing | $\{\emptyset\}$ | 0.8233 (0.064) | 0.2421 (0.579) | 0.4724 (0.072) |
|  |  | Selective | $\{\overset{\rightarrow}{\widetilde{X}}\}$ | **0.5260** (0.105) | **0.1497** (0.210) | **0.2633** (0.069) |
|  |  | Sel. Complement | $\{\widetilde{X}^{\rightarrow}\}$ | 0.8855 (0.056) | 0.4726 (0.971) | 0.6924 (0.152) |
| **PEHE$_{W=0}$** |  | All | $\{\widetilde{X}^{\rightarrow}, \overset{\rightarrow}{\widetilde{X}}\}$ | 0.8918 (0.107) | 1.6743 (1.218) | 0.5684 (0.138) |
|  |  | Nothing | $\{\emptyset\}$ | 0.7079 (0.087) | 0.2719 (0.340) | 0.4829 (0.116) |
|  |  | Selective | $\{\overset{\rightarrow}{\widetilde{X}}\}$ | **0.5754** (0.092) | **0.1539** (0.209) | **0.2699** (0.064) |
|  |  | Sel. Complement | $\{\widetilde{X}^{\rightarrow}\}$ | 0.8342 (0.105) | 0.3531 (0.604) | 0.7313 (0.200) |
| **PEHE$_{W=1}$** |  | All | $\{\widetilde{X}^{\rightarrow}, \overset{\rightarrow}{\widetilde{X}}\}$ | 0.8650 (0.074) | 2.0708 (0.709) | 0.5169 (0.107) |
|  |  | Nothing | $\{\emptyset\}$ | 0.8844 (0.080) | 0.2798 (0.706) | 0.4650 (0.070) |
|  |  | Selective | $\{\overset{\rightarrow}{\widetilde{X}}\}$ | **0.5035** (0.132) | **0.1482** (0.213) | **0.2590** (0.075) |
|  |  | Sel. Complement | $\{\widetilde{X}^{\rightarrow}\}$ | 0.9149 (0.068) | 0.5332 (1.197) | 0.6664 (0.157) |

## D    EXHAUSTIVE DAG-SEARCH OVER CIT AND CIO

CIT and CIO are assumptions over $X$, $\widetilde{X}$, $Z$, $Y(w)$, and $W$. While keeping $(X) \to (Z) \to (\widetilde{X})$ fixed, we discuss each other DAG that respects Eq. (2) or Eq. (3).

In particular, for a DAG to be a valid description for missingness in CATE, they *should contain*:
$(X) \to (Y)$, which is a standard arrow in CATE (cfr. Fig. 1a). Essentially, the potential outcome $Y(w)$ is the result of a natural process involving the fully observed $X$.
$(\widetilde{X}) \to (W)$, when a clinician determines treatment, they have to make due with what is given to them, i.e. the observed covariate set $\widetilde{X}$, which may be partially unobserved due to missingness. This proxy to $X$ is the best a clinician has available to them as the fully observed $X$ is not always available to them.
$(Z) \to (W)$, throughout our work we consider $Z$ to be different from $\widetilde{X}$, where $\widetilde{X}$ entails the actual *values* of the covariate set, $Z$ indicates their presence. If or not a variable is present for a clinician to base their treatment decision on, can have an effect on their eventual decision. Say that it is too risky for a particular treatment when a patient's blood-pressure is not observed, then a different treatment option will be chosen (or the variable will be measured before a decision is made). In this setting, the absence of a value has determined treatment, leading $Z$ into $W$.

Continuing our discussion, for a DAG to be valid as a missingness description, they *should not contain* the following:
$(X) \to (W)$, as, again, $X$ is simply not available for a clinician to base their treatment-decision on.
$(\widetilde{X}) \to (Y)$, as $Y(w)$ is the result of a natural process, it should depend on $X$, not $\widetilde{X}$.
$(Z) \to (Y)$, similarly, $Y(w)$, should not depend on the missingness $Z$ as different datasets on the same person would register different outcomes when outcomes depend directly on $Z$, which cannot happen.

**Permutations on CIT.** For CIT, there cannot be a direct arrow from $X$ to $W$ (a feat we agree with), as it automatically violates Eq. (2). As long as there exists a path between $W$ and $Y$, excluding the connecting SWIG-path [Richardson & Robins, 2013], we consider the DAG valid. When excluding paths *going up* from $W$ or $Y(w)$, the total amount of valid DAGs amounts to $(C_3^1 + C_3^2 + C_3^3) \cdot (C_2^1 + C_2^2) = 21$, where $C_3$ is coming from three potential paths into $Y$ (having three variables, other than $W$ and $Y(w)$), and $C_2$ is coming from two potential paths into $W$ (three variables, excluding $X$). All the CIT DAGs, including the one presented in Mayer et al. [2020b] are illustrated in Fig. 5 (the DAGs presented in [Mayer et al., 2020b] correspond to Figs. 5a and 6a for CIT and CIO, respectively).

Figure 5 is organised as follows: each two rows contain 7 graphs (corresponding to $(C_3^1 + C_3^2 + C_3^3)$) where we vary the arrows going into $Y(w)$; there are three sets of graphs (where a set is two rows), for each set we vary the arrows going into $W$ (where three then corresponds to $(C_2^1 + C_2^2)$).

Handy with Table 4, we find that Fig. 5g is the only valid DAG from all CIT (and later we see all CIO) compatible DAGs. Inspecting Fig. 5g we notice that it corresponds exactly with Fig. 1d, without the birdirectional arrow between $W$ and $Z$, which is a result of informative versus uninformative missingness. Splitting $Z$ in $Z^{\to}$ and $^{\to}Z$ results automatically in MCM.

**Permutations on CIO.** For CIO, the definition does not allow an arrow from $X$ to $Y(w)$, similarly to the definition of CIT not allowing an arrow from $X$ to $W$. Contrasting CIT, however, we find this restriction not sensible. As we have already argued above, $X$ is the *only* variable "justified" to influence the potential outcomes, $Y(w)$, directly. Any other variable influencing $Y(w)$ would result in contrasting outcomes over different datasets, which would imply, for example, different tumour sizes for the same person across different datasets. Having CIO and accompanying Eq. (3), results in 21 (applying the same calculation as for CIT) non-sensible DAGs, including the DAG presented in Mayer et al. [2020b]. All these DAGs are listed in Fig. 6, and evaluated (like for CIT) in Table 4.

Table 4: **Validity of CIT and CIO.** There are six criteria we argue missingness in CATE should follow. These criteria are indicated in the column headers as directed dependencies missingness should, or should not, include. Assuming that there are no arrows *going up* from $W$ and $Y(w)$ (for example, the potential outcomes $Y(w)$, cannot influence the covariates $X$), we have 42 DAGs that respect Eq. (1) and one of Eq. (2) or Eq. (3). Only one of these DAGs (indicated in green), respects each criteria. This DAG (Fig. 5g) is a permutation of CIT, and a version of MCM that does not assume factors, $Z^{\rightarrow}$ and $^{\rightarrow}Z$ in $Z$. In below table, "✓" means presence and "✗" means absence, when these icons are black absence or presence is positive, when they are gray they are negative.

| | | Does *not* contain | | | Does contain | | |
|---|---|---|---|---|---|---|---|
| | | $X \to W$ | $\tilde{X} \to Y$ | $Z \to Y$ | $X \to Y$ | $\tilde{X} \to W$ | $Z \to W$ |
| **CIT** | Fig. 5a | assum. | ✓ | ✓ | ✓ | ✓ | ✓ |
| | Fig. 5b | assum. | ✗ | ✓ | ✓ | ✓ | ✓ |
| | Fig. 5c | assum. | ✓ | ✗ | ✓ | ✓ | ✓ |
| | Fig. 5d | assum. | ✓ | ✓ | ✗ | ✓ | ✓ |
| | Fig. 5e | assum. | ✓ | ✗ | ✗ | ✓ | ✓ |
| | Fig. 5f | assum. | ✗ | ✓ | ✗ | ✓ | ✓ |
| | Fig. 5g | assum. | ✗ | ✗ | ✓ | ✓ | ✓ |
| | Fig. 5h | assum. | ✓ | ✓ | ✓ | ✗ | ✓ |
| | Fig. 5i | assum. | ✗ | ✓ | ✓ | ✗ | ✓ |
| | Fig. 5j | assum. | ✓ | ✗ | ✓ | ✗ | ✓ |
| | Fig. 5k | assum. | ✓ | ✓ | ✗ | ✗ | ✓ |
| | Fig. 5l | assum. | ✓ | ✗ | ✗ | ✗ | ✓ |
| | Fig. 5m | assum. | ✗ | ✓ | ✗ | ✗ | ✓ |
| | Fig. 5n | assum. | ✗ | ✗ | ✓ | ✗ | ✓ |
| | Fig. 5o | assum. | ✓ | ✓ | ✓ | ✓ | ✗ |
| | Fig. 5p | assum. | ✗ | ✓ | ✓ | ✓ | ✗ |
| | Fig. 5q | assum. | ✓ | ✗ | ✓ | ✓ | ✗ |
| | Fig. 5r | assum. | ✓ | ✓ | ✗ | ✓ | ✗ |
| | Fig. 5s | assum. | ✓ | ✗ | ✗ | ✓ | ✗ |
| | Fig. 5t | assum. | ✗ | ✓ | ✗ | ✓ | ✗ |
| | Fig. 5u | assum. | ✗ | ✗ | ✓ | ✓ | ✗ |
| **CIO** | Fig. 6a | ✓ | ✓ | ✓ | ✓ | ✓ | ✓ |
| | Fig. 6b | ✓ | ✓ | ✓ | ✓ | ✗ | ✓ |
| | Fig. 6c | ✓ | ✓ | ✓ | ✓ | ✓ | ✗ |
| | Fig. 6d | ✓ | ✓ | ✓ | ✗ | ✓ | ✓ |
| | Fig. 6e | ✓ | ✓ | ✓ | ✗ | ✓ | ✗ |
| | Fig. 6f | ✓ | ✓ | ✓ | ✗ | ✗ | ✓ |
| | Fig. 6g | ✓ | ✓ | ✓ | ✓ | ✗ | ✗ |
| | Fig. 6h | ✓ | ✗ | ✓ | ✓ | ✓ | ✓ |
| | Fig. 6i | ✓ | ✗ | ✓ | ✓ | ✗ | ✓ |
| | Fig. 6j | ✓ | ✗ | ✓ | ✓ | ✓ | ✗ |
| | Fig. 6k | ✓ | ✗ | ✓ | ✗ | ✓ | ✓ |
| | Fig. 6l | ✓ | ✗ | ✓ | ✗ | ✓ | ✗ |
| | Fig. 6m | ✓ | ✗ | ✓ | ✗ | ✗ | ✓ |
| | Fig. 6n | ✓ | ✗ | ✓ | ✓ | ✗ | ✗ |
| | Fig. 6o | ✓ | ✓ | ✗ | ✓ | ✓ | ✓ |
| | Fig. 6p | ✓ | ✓ | ✗ | ✓ | ✗ | ✓ |
| | Fig. 6q | ✓ | ✓ | ✗ | ✓ | ✓ | ✗ |
| | Fig. 6r | ✓ | ✓ | ✗ | ✗ | ✓ | ✓ |
| | Fig. 6s | ✓ | ✓ | ✗ | ✗ | ✓ | ✗ |
| | Fig. 6t | ✓ | ✓ | ✗ | ✗ | ✗ | ✓ |
| | Fig. 6u | ✓ | ✓ | ✗ | ✓ | ✗ | ✗ |

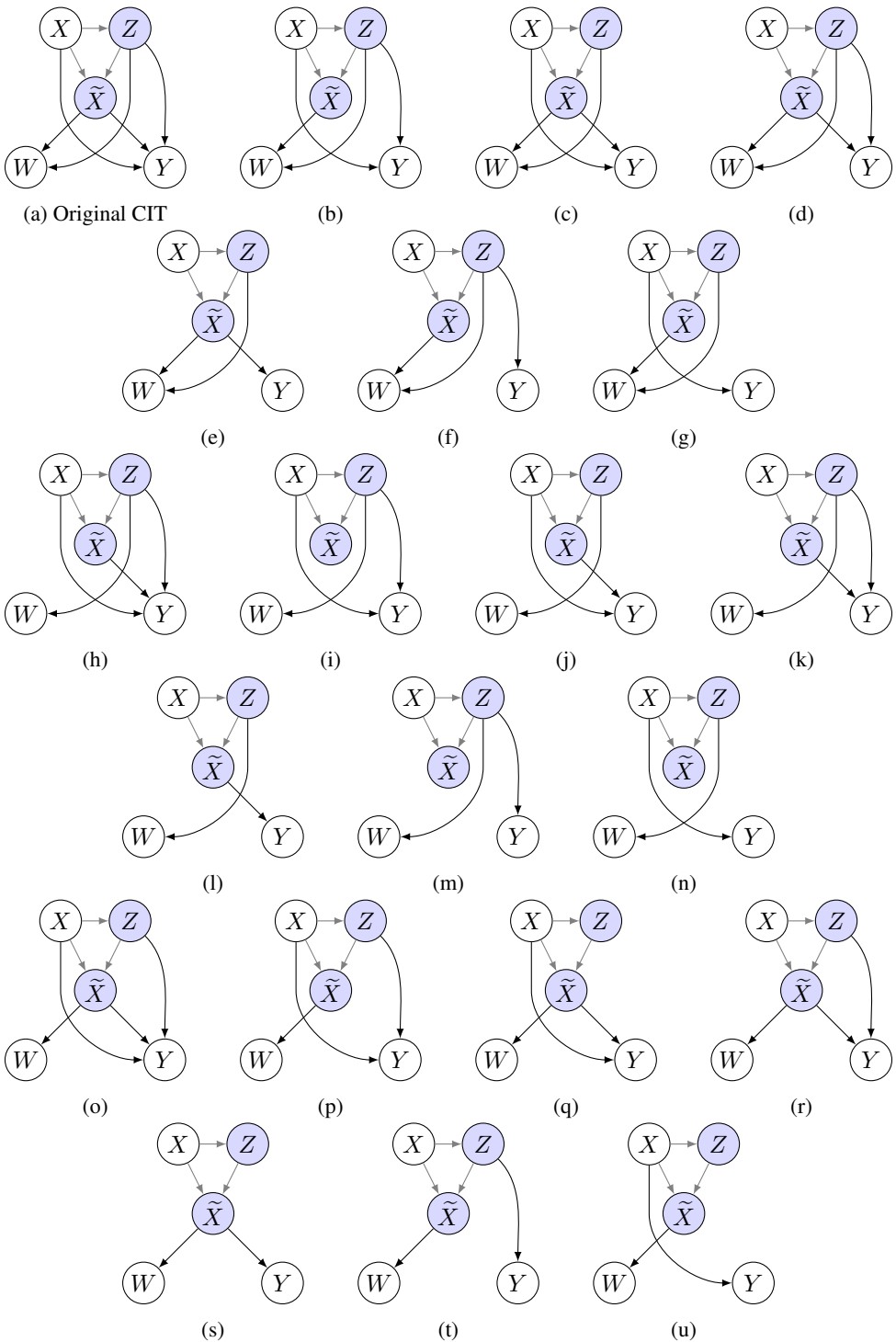

Figure 5: **Permutations on CIT.** There are a total of 21 DAGs that respect Eq. (1) and Eq. (2). From these DAGs, only one — Figure 5g — is acceptable. Add only the assumption that some elements in $Z$ are informative, and some elements in $Z$ are uninformative, and we automatically arrive at MCM. Note that we exclude edges *going up* from $W$ or $Y$ $(w)$, similarly to Richardson & Robins [2013].

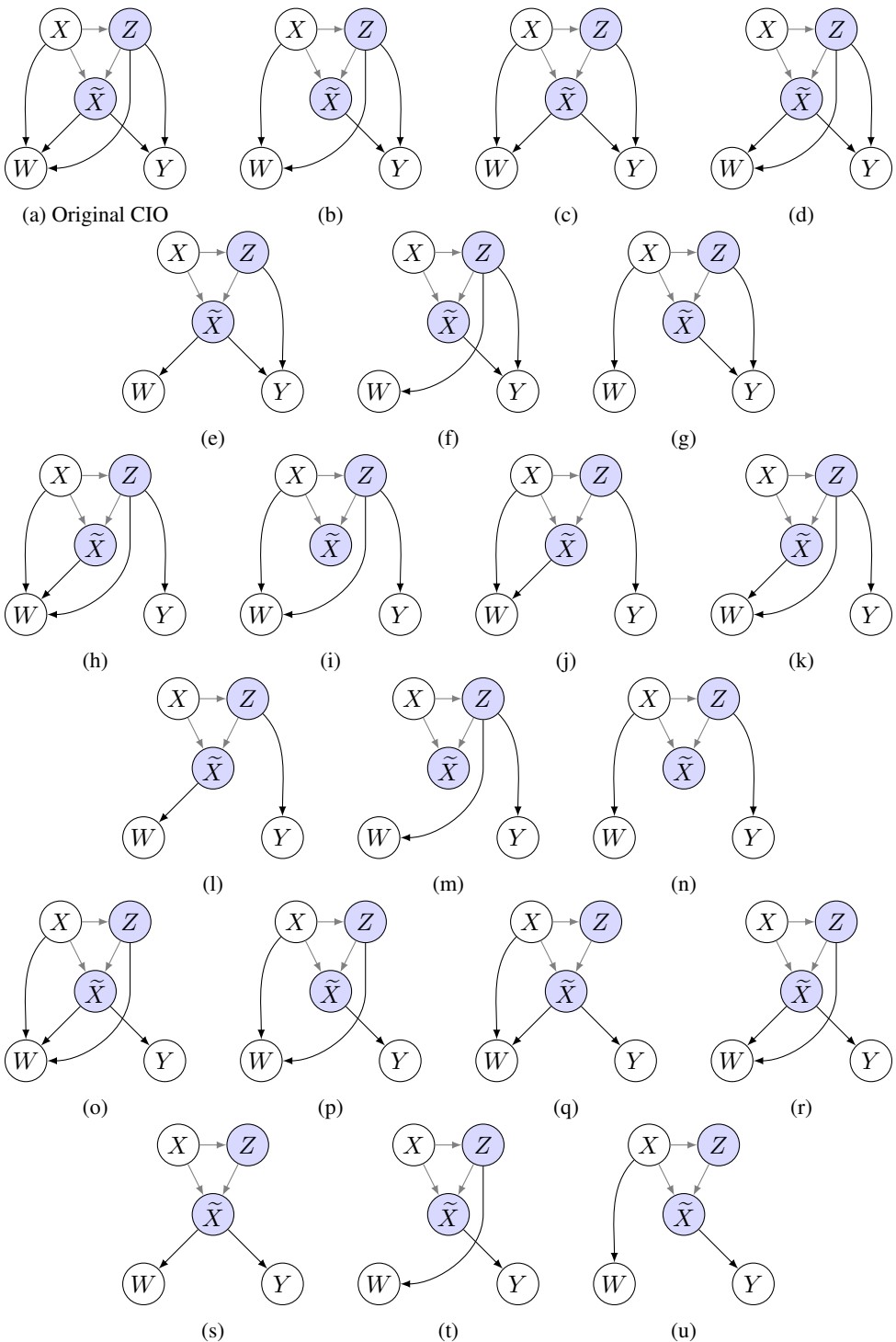

Figure 6: **Permutations on CIO.** There are 21 possible DAGs that respect Eq. (3). None of them is acceptable as they, by definition, cannot include a direct edge from $X$ to $Y(w)$. Having a direct edge between $X$ and $Y(w)$ encodes dependence, despite conditioning on $\widetilde{X}$ and $Z$. Note that we have excluded edges *going up* from $W$ or $Y(w)$, in similar fashion to Richardson & Robins [2013].

