# OpenReview forum: "To Impute or Not To Impute? Missing Data in Treatment Effect Estimation"
_ICLR.cc/2022/Conference — ICLR 2022 Submitted_

### Official Review · Reviewer_ZWmF · 2021-10-20

**Correctness:** 2
**Technical Novelty And Significance:** 3
**Empirical Novelty And Significance:** 2
**Recommendation:** 5
**Confidence:** 5

**Main Review:**

The questions they are trying to solve is an important problem.
For example in the discussion the author emphasize that "more care and thought should
be put into imputing missing data when estimating treatment effects." which I agree with completely.

The authors are using simplified DAG graphic tool to modeling the exposure, mediator, outcome variables with missingness and propose the causal relationship of missingness. In the real world data, one covariate may cause missingness of an exposure, which could be the predictor of the outcome variable. However, the missingness of this covariate may have no direct correlation with outcome variable. Therefore, to determine which variable will be imputed or not will not be simply determined by strategy will be limited.

Missingness in one variable may have direct or indirect relationship with missingness in the other variable or outcome variable. Alternative speaking, this correlation may not be linear. Which variables will be imputed or not imputed for the prediction of outcome variable will be  determined using machine learning approach for the feature selection.

Finally, even though it is difficult to use real word data for experimentation, it is important to attempt to do that. The imputation procedure has to be designed for a specific field as pattern of missingness may be very different among disciplines. The pattern of missingness and mechanism for a given dataset should be recognized. Whether the competition is favoring a certain method or procedure has to be determined in the “real-world” data with “real-world” missingness by considering recognized and unrecognized missing pattern/mechanism, as well as the plausible distribution of missing data. Testing a method on synthetic data, with no regards to observed patterns of missingness may only add noise to the field. A recent paper in the field was just published this month: https://www.nature.com/articles/s41746-021-00518-0

Therefore, even though the idea of improving imputation is excellent, the logic used in this study is not robust.

**Summary Of The Paper:**

"we identify a new missingness mechanism, which we term mixed confounded missingness (MCM), where some missingness determines treatment selection and other missingness is determined by treatment selection."

The author gave a new term called “MCM” which is not something new. MCM is a type of MNAR (Missing-Not-At-Random).
MCM cannot be listed parallel to MCAR, MAR and MNAR.



**Summary Of The Review:**

Even though the idea of improving imputation is excellent, the logic used in this study is not robust and does not take into account true realistic situations.

Use of only synthetic data, when we have access to many real-world datasets is not acceptable. How the synthetic data was created has a direct effect on how the methods will work. It is possible to use real-world data and introduce additional hold-out values to evaluate the methods. see for example: https://www.nature.com/articles/s41746-021-00518-0

The importance of real-world data is key here because the whole idea behind this paper is that missingness patterns are not completely understood.

---

> ### Author Response · Authors · 2021-11-16
> **Dear reviewer ZWmF**
>
> Dear reviewer ZWmF, thank you for reading and reviewing our manuscript. We also thank you for acknowledging that imputation for treatment effect estimation is indeed a very important problem. We address your comments point-by-point below.
>
> > [...] the missingness of [an outcome-predictor] covariate may have no direct correlation with outcome variable. Therefore, to determine which variable will be imputed or not will not be simply determined by strategy will be limited.
>
> In our paper, as well as the definitions of MNAR and MAR, missingness is a direct function of $X$ (in both ${}^{\to}Z$ and $Z^{\to}$). Similarly for outcome, which is also a function of $X$. Having these dependencies makes $X$ a confounder of $Z$ and $Y$, which correlates them. We are not sure which point you make above? However, we do wish to stress that the reason one may want to impute does not change based on whether or not they are correlated with the outcome variable. Instead, we need to impute (or refrain from imputing) based on the intermediate relationship (as a mediator or as a collider) between $X$ and $W$. It is the relationship between treatment and covariates we have to care about when making unbiased treatment effect estimates.
>
>
> > Missingness in one variable may have direct or indirect relationship with missingness in the other variable or outcome variable. Alternative speaking, this correlation may not be linear.
>
> You are correct. However, we never state that the missingness relationship should be linear. In fact, we remain faithful to the graphical literature on causality, where the only assumption on our edges is that they remain identifiable.
>
>
> > Which variables will be imputed or not imputed for the prediction of outcome variable will be determined using machine learning approach for the feature selection.
>
> Note that we **do not** present a method in this paper. We identify a major problem with methods currently in use. A strategy to build a novel imputation method, inspired by our theory, could indeed resort to feature selection.
>
> > Finally, even though it is difficult to use real world data for experimentation, it is important to attempt to do that.
>
> While we agree that it is important to use real world data where possible, as is stated throughout the treatment effects literature, it is not just difficult to use real data but **impossible**. Making an "attempt" to use real data is not at all helpful. Either real data can be used, and should be, or it cannot and therefore will (should) not be.

---

### Official Review · Reviewer_8bft · 2021-10-27

**Correctness:** 1
**Technical Novelty And Significance:** 1
**Empirical Novelty And Significance:** 1
**Recommendation:** 3
**Confidence:** 4

**Main Review:**

Authors use graphic causal models to model the various missingness in causal effect estimation. This is an interesting exercise.

However, I doubt the correctness of mapping Figure 1(d) to Figure 3. In Figure 1(d) the relationships between the rightmost four variables are "W \to Z_2 \to X'_2 \leftarrow X" (Note that I use Z_2 and X'_2 to replace the superscripted Z and X in Figure 1(d)). We should understand the relationships in the following way. W determines the missing values of Z_2 and subsequently the observed values of X'_2. This does not mean that W will determine X_2 when it does not include missing values. Using the authors' example, the participants of the job training program provide additional information whereas the non-participants do not. So, the missing values in X'_2 are determined by W. However, if we have a magical way to collect the additional information from non-participants (i.e. the missing values are imputed perfectly). The edge W \to real X_2 will be broken. Figure 1(d) becomes Figure 1(c). If X_2 is imputed correctly, there is not an edge W \to X_2 and the discission before "selective imputation" on Page 7 is invalid.

Given that Figure 3 is confusing and leads to an incorrect conclusion. There is a soundness problem with the paper.

Since the experiments follow Figure 3, the results cannot be trusted. If X_2 is a part of the covariate set, there is no edge W \to X_2 without missing values. The causal effect on the set of X is unbiased. If X_2 is imputed correctly, the causal effect is also unbiased.

Interestingly, there is not a discussion of how the missing values are imputed.


**Summary Of The Paper:**

This paper studies dealing with missing values in estimating treatment effects. Authors identify a new missingness mechanism, mixed confounded missingness (MCM), including missingness that determines treatment selection and missingness that is determined by treatment selection. The authors show that both imputation and no imputation lead to poor treatment effect estimations. The authors present a selective imputation strategy that informs which variables should be imputed and which should not. They empirically demonstrate the effectiveness of the strategy.


**Summary Of The Review:**

There is a soundness problem with the paper.

---

> ### Author Response · Authors · 2021-11-16
> **Dear reviewer 8bft**
>
>
> Dear reviewer 8bft, we thank you for reading our manuscript.  However, we believe that the points you make are incorrect and fundamentally disagree with the technical criticisms of our manuscript. Your review contains a number of misunderstandings, not only of our paper, or causality, but also probability.
> We address your comments point-by-point below.
>
> **Mapping fig. 1(d) to fig. 3.**
>
> Fig. 1(d) and fig. 3 are _equivalent_. We believe there are several misunderstandings in your review when you claim that fig. 1(d) and fig. 3 are not equivalent.
>
> > [...] [**point 1--**] We should understand the relationships in the following way. $W$ determines the missing values of $Z_2$ and subsequently the observed values of $X'_2$. This does not mean that $W$ will determine $X_2$ when it does not include missing values. [...] [**point 2--**] However, if we have a magical way to collect the [non-observed values] (i.e. the missing values are imputed perfectly). The edge $W \to \text{real} X_2$ will be broken. Figure 1(d) becomes Figure 1(c). If $X_2$ is imputed correctly, there is not an edge $W \to X_2$ and the discission before "selective imputation" on Page 7 is invalid.
>
>
> **point 1--** These edges **are not** individualistic in nature, they describe distributions. By reasoning about the situation "when there is no missingness", the reviewer is conditioning on the values of a given random variable. Essentially saying _"when we condition on what $X_2$ is, there is no longer influence from other variables on $X_2$"_. Your comment is equivalent to saying that for 2 random variables $X$ and $Y$ which are not independent (suppose they're jointly Gaussian with non-zero covariance if you like), that conditional on the event $X = x$, $X$ and $Y$ are independent. This is trivially true because, conditional on $X = x$, $X$ reduces to constant and is independent of (basically) anything. This holds for all $x$. This does not lead to the conclusion that $X$ and $Y$ are independent.
>
> **point 2--** As to your second point, this is precisely our claim - we need to impute $X_2$. We are very confused as to how this invalidates any of our discussion, because the entire purpose of the discussion is to identify the fact that imputing $X_2$ **is the right thing to do**.
>
>
> > Given that Figure 3 is confusing and leads to an incorrect conclusion. There is a soundness problem with the paper.
>
> There is no soundness problem with our paper. Not only does our theory hold for fig. 3, it actually also holds for fig. 1(d). The reason is simple: _they are equivalent_. In any dataset, $Z$ and $X$, are always taken together. They are considered "the object we observe", including missing values. The reason we split them up is to think about their influence separately, as is typically done in work concerning missingness. However, our point is made much clearer when collapsing them together, as together they form a textbook collider structure.
>
>
> > Since the experiments follow Figure 3, the results cannot be trusted. If $X_2$ is a part of the covariate set, there is no edge $W \to X_2$ without missing values. The causal effect on the set of $X$ is unbiased. If $X_2$ is imputed correctly, the causal effect is also unbiased.
>
> But we are precisely in the setting where, for some of the samples, there **are** missing values. Edges do not exist on a per-sample level but in the dataset as a whole, much like when talking about distributions that is across the whole dataset. As pointed out above, conditioning down to a single realisation of a random variable trivially causes independence, but does not imply independence between the unconditioned RVs at all.
>
> Furthermore, the generated data follow fig. 1(d), as we generate $Z$ and $X$ separately. We refer to our appendix, which includes the code that generates these data, as well as the actual python files and notebook to use this code. We believe both your theoretical point as well as your empirical one, is wrong.
>
>
> > Interestingly, there is not a discussion of how the missing values are imputed.
>
> The imputation method used is clearly stated on page 8 of our paper in the paragraph titled "Imputation". There, we state that we use MICE to perform imputation for the results in the paper.

---

> > ### Comment · Reviewer_8bft · 2021-11-21
> > **I still have doubts**
> >
> >
> > Let me make my doubts clear.
> > Claim one: “We argue that MCM is a general-purpose missingness mechanism”
> >
> > When the missing values are not a result of the treatment choice. Figure 1(d) is reduced to Figure 1(c). This means that M(N)AR is a special case of MCM. This is good.
> >
> > Let us look at Figure 3, \rightarrow {\hat X} (observed) is only determined by \rightarrow {\tilde X}.
> >
> > I have the following doubt.
> >
> > Assume there are is no missingness as a result of the treatment choice. Or there is no edge between W and \rightarrow {\tilde X}. However, there is missingness by M(N)AR. Can missing values by M(N)AR be estimated using the causal graph in Figure 3? It seems not since there is not a causal path between Z\rightarrow (missingness by M(N)AR) and \rightarrow {\hat X} observed values.
> >
> > Therefore, Figure 3 does not generalise the M(N)AR mechanism.
> >
> >
> > Claim 2: “We propose a strategy to handling missing data in treatment effects, termed selective imputation.”
> >
> > I do not find the strategy in Section 4. I do find text that “a good imputation strategy should aim to make X independent of the treatment, such that ignorability may hold.” As far as I know, the ignorability cannot be tested in data. In data, we do not know whether the ignorability is satisfied or not. The statement a good imputation strategy is not rigorous.

---

> > > ### Author Response · Authors · 2021-11-22
> > > **Dear reviewer 8bft**
> > >
> > > Dear reviewer 8bft,
> > >
> > > Thank you for responding. Please consider the below, where we clarify further, in aim to resolve your remaining concerns.
> > >
> > > First, we will reiterate the purpose of fig. 1(d) and fig. 3. Fig. 1(d) depicts the missingness mechanism MCM, a special case of M(N)AR (which, from your comment, you seem to agree with). Fig. 3 depicts the missingness mechanism MCM together with our proposed strategy on how to impute when in an MCM-setting. This leads to the addition of ${}^{\to}\dot{X}$ in fig. 3 compared with fig. 1(d). The only other difference between fig. 1(d) and fig. 3 is the simplification of graphical depiction by collapsing $Z$ into $\widetilde{X}$. By simplifying fig. 1(d) into the mediator/collider structure in fig. 3, we can clearly argue why one should only impute ${}^{\to}\widetilde{X}$.
> > >
> > > **Claim 1.**
> > > >Assume there are is no missingness as a result of the treatment choice. Or there is no edge between W and \rightarrow {\tilde X}. However, there is missingness by M(N)AR. Can missing values by M(N)AR be estimated using the causal graph in Figure 3? It seems not since there is not a causal path between Z\rightarrow (missingness by M(N)AR) and \rightarrow {\hat X} observed values.
> > >
> > > When there is no missingness as a result of the treatment choice, MCM is transformed into M(N)AR, as agreed by the reviewer in their comment stating fig. 1(d) is reduced to fig. 1(c). This can be achieved not by simply removing the arrow, but by decreasing the size of the partition, ${}^{\to}Z$, to $0$. Essentially, in the M(N)AR setting, the partitions have the following sizes: $|{}^{\to}Z|=0$ and $|Z^\to| = |X|$.
> > > Since $|{}^{\to}{\widetilde X}| = 0$, in the M(N)AR setting there is nothing to impute according to the imputation strategy depicted in fig. 3. Furthermore, since $|{}^{\to}{Z}| = 0$ and $|{}^{\to}{\widetilde X}| = 0$, the causal path that the reviewer believes is missing cannot possibly exist.
> > >
> > > >Therefore, Figure 3 does not generalise the M(N)AR mechanism.
> > >
> > > As discussed above, when there is no missingness as a result of treatment, the missingness mechanism of MCM included in fig 3. becomes equivalent to M(N)AR. Thus, MCM does generalize M(N)AR.
> > >
> > >
> > > **Claim 2.**
> > > Indeed, one cannot test for ignorability between the potential outcomes and treatment, since we never observe the counterfactual outcome. This is not new, nor is it surprising. The treatment effects community has dealt with this issue by assuming a certain structure of the data (which is displayed in fig. 1 (a) and also defined in Assumption 2). Most treatment effects methods assume ignorability, despite the inability to test this in real data. In our paper, we introduce a way in which that same community can assume structure when there is missingness in the data.
> > >
> > > In Theorem 1 we state exactly what a good imputation method needs to provide such that, given the presented structure, we make unbiased estimates. We then prove this statement in our appendix. We believe this theoretical contribution makes our statement rigorous. We would be grateful if the reviewer could let us know what would make our statement rigorous beyond this.
> > >
> > > Once more, we truly appreciate getting back to us. We are eager to engage and discuss with you further should there still be items that remain unclear.

---

> > > > ### Comment · Reviewer_8bft · 2021-11-23
> > > > **Re authors**
> > > >
> > > > Thanks for the explanations. There are some gaps between our understandings.
> > > >
> > > > >> …, in the M(N)AR setting there is nothing to impute according to the imputation strategy depicted in fig. 3.
> > > >
> > > > This contradicts what M(N)AR mechanism in Figure 1(c) suggests. In Figure 1(c), the variable that blocks the backdoor path for estimating the causal effect of $W$ on $Y$ is X (without missing values, or imputed). $\widetilde {X}$ does not block all backdoor paths. Therefore, there needs imputation for unbiased causal effect estimation.
> > > >
> > > > This is what I mean by that Figure 3 does not generalise the M(N)AR mechanism.
> > > >
> > > >
> > > > >>Claim 2.
> > > > >>Indeed, one cannot test for ignorability between the potential outcomes and treatment, since we never observe the counterfactual outcome. This is not new, nor is it surprising. The treatment effects community has dealt with this issue by assuming a certain structure of the data (which is displayed in fig. 1 (a) and also defined in Assumption 2). Most treatment effects methods assume ignorability, despite the inability to test this in real data. In our paper, we introduce a way in which that same community can assume structure when there is missingness in the data.
> > > > >>In Theorem 1 we state exactly what a good imputation method needs to provide such that, given the presented structure, we make unbiased estimates. We then prove this statement in our appendix. We believe this theoretical contribution makes our statement rigorous. We would be grateful if the reviewer could let us know what would make our statement rigorous beyond this.
> > > >
> > > > The authors have been aware that it does not make the ignorability hold to make X independent of the treatment in data, and hence use “may hold”. The statement is uncertain. So, "a good imputation strategy should aim to make )_X independent of the treatment, such that ignorability may hold." is not a sound strategy.
> > > >
> > > > Furthermore, Figure 3 misses an edge from $\widetilde{X}{}^{\to}$ to ${}^{\to}\dot{X}$. When there are missing values from both M(N)AR and MCM, $\widetilde{X}{}^{\to}$ affects imputed values too and hence there is such an edge. In data, we only observe missing values and do not know whether they are from M(N)AR or MCM and hence cannot impute missing values from one mechanism only. Based on the new DAG, Theorem 1 does not hold anymore. This explains the inconsistency between my conclusion and the authors' conclusion in the previous discussions (Claim 1).
> > > >
> > > > If the authors consider that there are missing values from MCM only, the problem discussed is very restrictive.

---

> > > > > ### Author Response · Authors · 2021-11-24
> > > > > **Dear reviewer 8bft**
> > > > >
> > > > > Thank you again for responding.
> > > > >
> > > > > We believe we now understand your confusion. Given only the graphical structure in Fig 1(c) and no further knowledge of what the variables actually depict nor what form the arrows take, you are correct in stating that $\tilde{X}$ alone does not block the backdoor paths between $W$ and $Y$. However, for the actual problem being considered, in which $\tilde{X}$ represents the features with missingness and $Z$ represents the missingness indicator, $Z$ is observed whenever $\tilde{X}$ is. The pair $(\tilde{X}, Z)$ certainly do block all backdoor paths. It should also be stressed that performing imputation does not simply allow you to obtain $X$. The imputation strategy we suggest specifically targets independence from $W$. Any imputation strategy that takes as input only $\tilde{X}$ and $Z$ cannot hope to contain any information from $X$ that is not contained in $\tilde{X}$ and $Z$.
> > > > >
> > > > > As for the statement regarding ignorability; "such that ignorability may hold" does not imply it might not, it implies that _it is now able to hold_. Our statement is not meant to imply that it _can hold_, but rather that it _does hold_. We agree that the language would be clearer if we used the statement "such that ignorability holds", and will change it as such in our manuscript. With our theorem we– unambiguously –mean that independence between $W$ and $\dot{X}$ is sufficient to make ignorability hold.
> > > > >
> > > > > For the final statement regarding $\dot{X}$, it should be stressed that $\dot{X}$ _is our own construction_. It also does not make sense to talk about missingness from M(N)AR and MCM, as MCM _does generalise_ M(N)AR so any missingness from M(N)AR can be formulated using the same graphical model as MCM. Whether or not $\tilde{X}^{\rightarrow}$ directly affects the imputation is completely controlled _by us_ as $\dot{X}$ is _our own construction_.
> > > > >
> > > > > It is not clear to us at all what you mean by missing values that are from MCM only. Our missingness mechanism is _very clearly_ defined using the graphical model presented in Fig. 1(d) and the graph for M(N)AR is _very clearly_ a subgraph of MCM. Any missingness that can be described as M(N)AR can be described by MCM. Furthermore, a missingness mechanism depicts the reason _why_ a variable is missing. In both MCM and M(N)AR, missingness is caused by $X$. Additionally, MCM also considers missingness that is caused by the treatment. If, as you state, "there are missing values from both M(N)AR and MCM", then missingness can be described entirely by MCM.

---

### Official Review · Reviewer_efJq · 2021-11-04

**Correctness:** 3
**Technical Novelty And Significance:** 2
**Empirical Novelty And Significance:** 2
**Recommendation:** 3
**Confidence:** 3

**Main Review:**

**Strengths of paper**

- Paper presents a very elaborate writeup with a detailed introduction to causality and missingness in treatment effect estimation which makes it a good read for audience who are new to this field.

- Problem addressed is a relevant problem and is applicable across several real-world problems.

**Weakness**

- The paper has a lot of content on introducing causality which instead could have been used to add more use cases on how selective imputation can be useful for TE
- It does appear that the authors could have studied the exhaustiveness claim of MCM in depth to tease out the contributions of this paper better. This is a very broad claim with insufficient justification in the current draft.
- It is not exactly clear how this paper advances the technical state-of-the-art to clear the ICLR bar of acceptance. The general writeup and contributions make it a better fit for a venue like UAI and other similar avenues.

Overall, I would like the authors to think more broadly on the imputation problem itself as industry problems rarely rely on using any kind of imputation method (mainly due to skepticism and avoiding corrupting the data). It might help if authors can talk about broader adoption for this work for such a real-world setting.

**Summary Of The Paper:**

In this paper, the authors mainly study the problem of missing data in treatment effect estimation and highlight the importance of addressing this problem. The authors propose a selective imputation scheme which is more well suited for addressing missingness in such scenarios. Authors also present several sample scenarios illustratively which indicate potential issues with current methods while doing TE with missingness.  Empirical results compare different scenarios where general imputation schemes (imputing all data, no data, wrong data) can be much worse than their proposed method.

**Summary Of The Review:**

I have highlighted my major observations from this paper above. While the paper is written in a very elaborate manner, I do believe it is still not clear if the technical contributions are advancing the state-of-the-art to clear the ICLR bar of acceptance. Authors should tease out the technical contributions around exhaustiveness and others more clearly and remove big sections on assumptions, metrics (can be written briefly with citations). I would also recommend trimming down the graphs to only keep the most relevant DAGs with a clear messaging.

---

> ### Author Response · Authors · 2021-11-16
> **Dear reviewer efJq (part 1)**
>
> Dear reviewer efJq, thank you for reading our manuscript and providing a review of our paper.
>
>
> We believe that our paper identifies a **major problem** in the current state of treatment effects estimation– both in academia, as well as industry. In fact, your final comment,
> > [...] industry problems rarely rely on using any kind of imputation method (mainly due to skepticism and avoiding corrupting the data),
>
> completely justifies why our paper is of such importance. Namely, our paper shows (theoretically and empirically) that both imputing and not imputing "corrupts" the model by introducing bias.
>
> The treatment effects literature is mostly applied in the medical domain (e.g. in clinical trials [1-10], decision support [1, 10-13], and even in clinical policy [14-20]). In our paper we show, theoretically as well as empirically, that no imputation will lead to biased causal effect estimates. Ultimately, we argue that imputation requires careful thinking when estimating causal effects, as the naive solutions (imputing everything or imputing nothing) may lead to wrong conclusions in these critical scenarios.
>
>
> While we will respond to each of your comments below, we hope that the above at least clarifies that our paper does indeed push, and more importantly _questions_, the current state-of-the-art.
>
> ### Individual points
>
> > The paper has a lot of content on introducing causality which instead could have been used to add more use cases on how selective imputation can be useful for TE.
>
> There are two points hidden in the above: (i) the amount of causal introduction, (ii) the lack of use cases to illustrate why MCM (and selective imputation) matters. We shall address them separately:
>
> (i) Causality is central to MCM, treatment effects, and our argument why MCM is detrimental when not appropriately taken care of. Not only do we bring together two alternative views on causality (potential outcomes and graphical causality), each element introduced is required in our exposition later in the paper.
>
> (ii) Our paper highlights that *in general* selective imputation needs to be considered whenever the goal is to estimate treatment effects in the presence of missing data. That the missingness mechanism is *not* of the form we give (i.e. treatment depends on some missingness and causes other missingness) is something that needs to be verified in *any* setting with missingness and TE estimation.
>
> To motivate the necessity of this verification, we provide some example scenarios in which it certainly is not possible to ignore imputation, such as in a job program. Another such example is whether or not a patient should undergo surgery, where the treatment (the surgery) could bring to light additional information about the patient. Another, is ICU admission, where when in the ICU, a patient may be monitored much more carefully (i.e. collecting additional/other variables). In all these cases, the treatment may warrant additional data-collection, and the treatment itself may be determined based on incomplete data. These are not rare, or niche, scenarios.

---

> ### Author Response · Authors · 2021-11-16
> **Dear reviewer efJq (part 2)**
>
>
> > It does appear that the authors could have studied the exhaustiveness claim of MCM in depth to tease out the contributions of this paper better. This is a very broad claim with insufficient justification in the current draft.
>
> We would like to respond to this comment in two parts:
>
> 1. When considering a DAG, one is considering the assumptions made on the data's distribution. Using a causal structure, we can determine whether or not the causal effect is identifiable. Having 5 variables ($X$, ${}^{\to}Z$, $Z^{\to}$, $W$, $Y$), with $X$ and $Y$ as start and end-nodes respectively, one only has so many possible DAGs. With exhaustiveness, we simply mean that all reasonable scenarios (i.e. assumptions on $p(X, {}^{\to}Z, Z^{\to}, W, Y)$) are considered. For this, we check the "capacity" of the DAG and subsequently rule out the edges that do not make sense. While scenarios with hidden confounding or dependent partitions in Z may exist, they remain outside the scope of our paper (and most of the treatment effects literature), and are thus not considered. Note, however, that we state this in our "preliminaries" section (on p. 3).
>
>
> 2. We would also like to point out that even if MCM is indeed not exhaustive (i.e. covers all possible situations in the DAGs capacity), our claims on bias when fully imputing or not imputing still hold. There need only be the structure of treatment causing and caused by missingness for the resulting estimands to be biased. Thus, while we believe MCM does indeed cover all possible scenarios following our initial assumptions, this is only one of the contributions and does not directly affect the central contribution of our manuscript - identifying a source of bias, that was not previously known or considered, for treatment effect estimation when there is missing data.
>
> > It is not exactly clear how this paper advances the technical state-of-the-art to clear the ICLR bar of acceptance. The general writeup and contributions make it a better fit for a venue like UAI and other similar avenues.
>
> Treatment effects estimation, both ATE and CATE, are widely adopted in (as well as outside, cfr. our additional references) the machine learning community. Exposing a major problem in the current pipeline when adopting these methods not only advances our understanding (and thus the "state-of-the-art"), but also indicates where we as a community need to spend time and effort to advance our field even further. An academic conference on ML (such as UAI or ICLR) is the right venue for us to share our work.

---

> ### Author Response · Authors · 2021-11-16
> **Dear reviewer efJq (part 3 - additional references)**
>
>
> [1] Angrist, Joshua, and Guido Imbens. "Identification and estimation of local average treatment effects." (1995).
>
> [2] Gail, M. H., Wai-Yuan Tan, and Steven Piantadosi. "Tests for no treatment effect in randomized clinical trials." Biometrika 75.1 (1988): 57-64.
>
> [3] Wang, Sue‐Jane, Robert T. O'Neill, and HM James Hung. "Approaches to evaluation of treatment effect in randomized clinical trials with genomic subset." Pharmaceutical Statistics: The Journal of Applied Statistics in the Pharmaceutical Industry 6.3 (2007): 227-244.
>
> [4] Akobeng, A. K. "Understanding measures of treatment effect in clinical trials." Archives of disease in childhood 90.1 (2005): 54-56.
>
> [5] Jennison, Christopher, and Bruce W. Turnbull. "Mid‐course sample size modification in clinical trials based on the observed treatment effect." Statistics in Medicine 22.6 (2003): 971-993.
> APA
>
> [6] Case, L. Douglas, et al. "Interpreting measures of treatment effect in cancer clinical trials." The oncologist 7.3 (2002): 181-187.
>
> [7] Royston, Patrick, and Mahesh KB Parmar. "The use of restricted mean survival time to estimate the treatment effect in randomized clinical trials when the proportional hazards assumption is in doubt." Statistics in medicine 30.19 (2011): 2409-2421.
>
> [8] Brumback, Babette A. "A note on using the estimated versus the known propensity score to estimate the average treatment effect." Statistics \& Probability Letters 79.4 (2009): 537-542.
>
> [9] DerSimonian, Rebecca, and Nan Laird. "Meta-analysis in clinical trials." Controlled clinical trials 7.3 (1986): 177-188.
>
>
> [10] Adelman, Leonard. "Experiments, quasi-experiments, and case studies: A review of empirical methods for evaluating decision support systems." IEEE transactions on systems, man, and cybernetics 21.2 (1991): 293-301.
>
> [11] Yu, Genghua, et al. "Medical decision support system for cancer treatment in precision medicine in developing countries." Expert Systems with Applications 186 (2021): 115725.
>
> [12] Schulam, Peter, and Suchi Saria. "Reliable decision support using counterfactual models." Advances in Neural Information Processing Systems 30 (2017): 1697-1708.
>
> [13] Bennett, Casey C., Thomas W. Doub, and Rebecca Selove. "EHRs connect research and practice: Where predictive modeling, artificial intelligence, and clinical decision support intersect." Health Policy and Technology 1.2 (2012): 105-114.
>
>
> [14] Jordà, Òscar, and Alan M. Taylor. "The time for austerity: estimating the average treatment effect of fiscal policy." The Economic Journal 126.590 (2016): 219-255.
>
> [15] Wang, Aolin, Roch A. Nianogo, and Onyebuchi A. Arah. "G-computation of average treatment effects on the treated and the untreated." BMC medical research methodology 17.1 (2017): 1-5.
>
> [16] Hartman, Erin, et al. "From sample average treatment effect to population average treatment effect on the treated: combining experimental with observational studies to estimate population treatment effects." Journal of the Royal Statistical Society. Series A (Statistics in Society) (2015): 757-778.
>
> [17] Heckman, James J., and Edward Vytlacil. "Policy-relevant treatment effects." American Economic Review 91.2 (2001): 107-111.
>
> [18] Johnson, Lorraine, Mira Shapiro, and Jennifer Mankoff. "Removing the mask of average treatment effects in chronic Lyme disease research using Big Data and subgroup analysis." Healthcare. Vol. 6. No. 4. Multidisciplinary Digital Publishing Institute, 2018.
>
> [19] Kreif, Noémi, et al. "Regression-adjusted matching and double-robust methods for estimating average treatment effects in health economic evaluation." Health Services and Outcomes Research Methodology 13.2 (2013): 174-202.
>
> [20] Balzer, Laura B., et al. "Targeted estimation and inference for the sample average treatment effect in trials with and without pair‐matching." Statistics in medicine 35.21 (2016): 3717-3732.

---

### Decision · Program_Chairs · 2022-01-20

**Decision:**

Reject

**Comment:**

In this paper, the authors propose a new type of (missing not at random) model they call the MCM (mixed confounded missingness).
The authors further discuss that given their model, naive imputation strategies do not work, and a model-tailored imputation strategy is needed.

The reviewers did not receive the paper favorably, with main complaints centering around: (a) outlining novelty compared to existing approaches to missing data, (b) whether imputation is a good strategy for dealing with missing data, and (c) whether the paper's results are actually sound.

Here's my perspective on these worries.

The paper aims to deal with missing data in a causal inference context (in other words, the target of inference is a causal effect, and our data happens to have entries missing not at random).  Further, the paper aims to work within a graphical modeling formalism for missing data models.  Finally, the paper points out that imputation is to be done with care if data is missing not at random (a point both myself, and reviewers agreed with).

Areas of improvement in the paper, in my mind, would be: (i) better literature review and putting authors' work in context of prior work, (ii) being clear about identification, and (iii) discussion of estimation strategies (not just imputation).

Dealing with missing data (in particular right censoring, but also more general types of missingness) in causal inference is a very old problem, with an established literature in statistics and public health.  In fact, methods for dealing with both causal inference and missing data together are a part of standard graduate curriculum in epidemiology and biostatistics in many Universities.

(i) Literature review and context.  Some papers the authors may find helpful to review:

James M. Robins, Andrea Rotnitzky, Daniel O. Scharfstein.  Sensitivity Analysis for Selection bias and unmeasured Confounding in missing Data and Causal inference models.  Part of the The IMA Volumes in Mathematics and its Applications book series (IMA, volume 116).

This paper discusses lots of relevant things, but in particular sensitivity analysis methods to violations of MAR in settings the authors worry about.

James M. Robins. Non-response models for the analysis of non-monotone non-ignorable missing data. Statistics in Medicine, 16:21–37, 1997.

This paper is an early example of an MNAR model that may be represented by a directed acyclic graph.

Karthika Mohan, Judea Pearl, and Jin Tian. Graphical models for inference with missing data. In C.J.C. Burges, L. Bottou, M. Welling, Z. Ghahramani, and K.Q. Weinberger, editors, Advances in Neural Information Processing Systems 26, pages 1277–1285. Curran Associates, Inc., 2013.

Ilya Shpitser, Karthika Mohan, Judea Pearl.  Missing data as a causal and probabilistic problem.  In Proceedings of the Thirty First Conference on Uncertainty in Artificial Intelligence (UAI-15), pp. 802-811, AUAI Press, 2015.

Rohit Bhattacharya, Razieh Nabi and Ilya Shpitser. “Full Law Identification In Graphical Models Of Missing Data: Completeness Results.” In Proceedings of the Thirty-Seventh International Conference on Machine Learning (ICML-20), pp. 7153-7163, 2020.

These papers deal with general models of missing data using graphs.

Since the authors use graphical models as well, I urge them to put their contribution in context with this prior work.

(ii) Identification.  The authors should clearly discuss whether treatment effects are identified under their model, and if so, by what function.  If this function is not closed form (which can happen in missing data), this should be discussed as well.  This should be contrasted with other missing data work that derives identification under MNAR, particularly using graphs.

(iii) Estimation.  The authors chose to use imputation.  Imputation is a sampling approach to inference in missing data.  Others include maximum likelihood or Bayesian methods (via EM), or semi-parametric inference via influence functions.  If the authors chose to concentrate on imputation, specifically, they should explain why (as other methods have noted advantaged, e.g. statistical efficiency, quantification of uncertainty, etc.).

Cautioning against naive imputation is a fine thing to do, but everyone working on missing data problems already knows naive imputation does not work for MNAR data.  Please do not oversell your contributions.  Saying things like: "MCM being the first formalisation of a missingness mechanism when there are treatments at play." is neither true, nor helpful for the peer review process.

With all that said, the MCM model has the potential to be an interesting MNAR model, and placed in proper context of existing work, could be a very interesting addition to the missing data literature.  However, the draft needs a bit more work before it is ready for publication.